ecology, palaeontology

biological pump, Cambrian, computational fluid dynamics, *Isoxys*, pelagic

**Author for correspondence:**
Stephen Pates
e-mail: sp587@cam.ac.uk

# Vertically migrating *Isoxys* and the early Cambrian biological pump

Stephen Pates[1,2], Allison C. Daley[3], David A. Legg[4] and Imran A. Rahman[5]

[1]Museum of Comparative Zoology and Department of Organismic and Evolutionary Biology, Harvard University, Cambridge, MA, USA
[2]Department of Zoology, University of Cambridge, Cambridge, UK
[3]ISTE, University of Lausanne, Lausanne, Vaud, Switzerland
[4]Faculty of Science and Engineering, University of Manchester, Manchester, UK
[5]Oxford University Museum of Natural History, University of Oxford, Oxford, UK

SP, 0000-0001-8063-9469; ACD, 0000-0001-5369-5879; IAR, 0000-0001-6598-6534

The biological pump is crucial for transporting nutrients fixed by surface-dwelling primary producers to demersal animal communities. Indeed, the establishment of an efficient biological pump was likely a key factor enabling the diversification of animals over 500 Myr ago during the Cambrian explosion. The modern biological pump operates through two main vectors: the passive sinking of aggregates of organic matter, and the active vertical migration of animals. The coevolution of eukaryotes and sinking aggregates is well understood for the Proterozoic and Cambrian; however, little attention has been paid to the establishment of the vertical migration of animals. Here we investigate the morphological variation and hydrodynamic performance of the Cambrian euarthropod *Isoxys*. We combine elliptical Fourier analysis of carapace shape with computational fluid dynamics simulations to demonstrate that *Isoxys* species likely occupied a variety of niches in Cambrian oceans, including vertical migrants, providing the first quantitative evidence that some Cambrian animals were adapted for vertical movement in the water column. Vertical migration was one of several early Cambrian metazoan innovations that led to the biological pump taking on a modern-style architecture over 500 Myr ago.

## 1. Introduction

The biological pump is the process by which organic nutrients are transported from shallow ocean to deep sea [1]. Today this consists of two major vectors: passive sinking of organic matter aggregates and vertical movement of animals [1] (figure 1). The biological pump, the main driver of the marine carbon cycle, is responsible for approximately two-thirds of the vertical gradient of carbon in the ocean [2]. While transport of carbon to the deep ocean through vertical mixing of dissolved organic carbon (DOC) carries significant amounts of carbon to depth, 95% of DOC cannot be used as food by marine organisms [1,3]. By contrast, aggregates and vertical migrants concentrate organic matter in a form that can be used by demersal animals [1] and are therefore crucial for the establishment and sustenance of deep-water communities.

The biological pump was very different before the Cryogenian Period (more than 720 Ma). Cyanobacterial picoplankton (0.2–2.0 µm) domination led to a stratified and turbid water column (e.g. [4–6]). Cells were too small to sink in significant numbers, so most nutrients were recycled within the surface waters, with little export to the deeper ocean [4–6]. The flux of the aggregation vector (figure 1) would have increased when primary productivity shifted to be eukaryote-dominated during the Cryogenian ∼ 650 Ma [7], and further when Cambrian suspension feeding sponges and pelagic phytoplanktivores applied a size-selective pressure for larger primary producers [8–12]. The carcasses, sloppy feeding and faecal pellets of macrozooplankton, including phytoplanktivorous

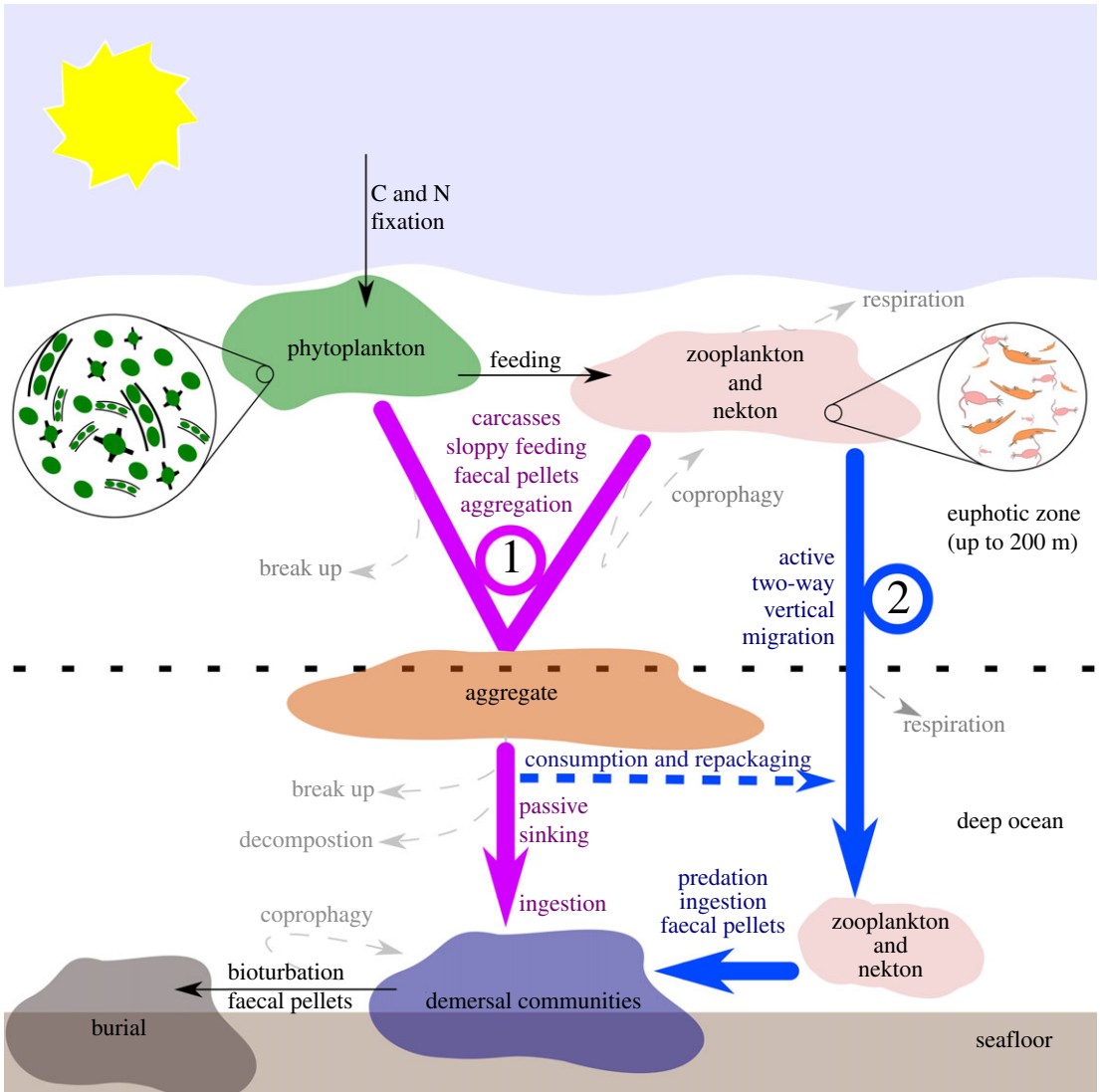

**Figure 1.** Simplified architecture of the modern biological pump. Arrows trace pathway of carbon (C) and nitrogen (N) and other nutrients. Numbers indicate two main biologically mediated vectors that transport nutrients fixed by phytoplankton from the surface ocean to demersal communities. (1) Aggregation vector of phytoplankton, faecal pellets and other organic matter which sinks passively through the water column. (2) Vertical migration vector driven by active two-way migration by metazoans. (Online version in colour.)

crown-group crustaceans (e.g. [9]), would have further increased the formation and size of aggregates, and thus the flux along this vector. However, despite the importance of vertical migration in the transport of nutrients necessary to sustain mesopelagic and deep-water ecosystems in the modern ocean (e.g. [13]), this vector is poorly understood in the Cambrian.

Comparisons of the morphology of Cambrian fossil organisms with modern vertically mobile pelagic animals provides the opportunity to infer whether vertical migration occurred in oceans over 500 Ma. Qualitative comparisons between the pelagic crustacean *Gnathophausia* and the nektonic stem group euarthropod [14] *Isoxys* (e.g. [15,16]) suggest the latter is a promising candidate. The 20 *Isoxys* species so far formally described are united by the presence of a bivalved carapace which bears both anterior and posterior spines [17] (figure 2), and the genus has a cosmopolitan distribution (e.g. [18]). The presence of eyes that can comprise approximately 10% of the body length, a digestive tract with paired serial midgut glands, and a pair of anteriorly positioned raptorial appendages (figure 2) support a predatory habit for *Isoxys*, which would have been well suited for capturing small soft-bodied invertebrates [19]. *Isoxys* is unusual for Cambrian animals as a pelagic lifestyle has been proposed (e.g. [18]), although some recent workers have suggested a potential hyperbenthic lifestyle (1–10 m above the bottom), with individuals capable of moving small distances vertically [19,20]. However, while *Isoxys* carapaces appear to show adaptations for hydrodynamic streamlining, interspecific differences in both carapace asymmetry and spine lengths (e.g. figure 2), as well as soft parts, suggest that different species may have occupied distinct niches, including some much closer to the seafloor [21].

Here, we provide the first quantitative assessment of carapace shape variation across the group and compare *Isoxys* to other Cambrian 'bivalved' euarthropods and *Gnathophausia*. Subsequently, through computational fluid dynamics (CFD) simulations, we test the importance of the spines and carapace outline for generating lift and reducing drag, and thus the ability of different taxa to move vertically through the water column. These analyses support the hypothesis that *Isoxys* taxa occupied a variety of distinct niches in Cambrian oceans, including vertical migrants.

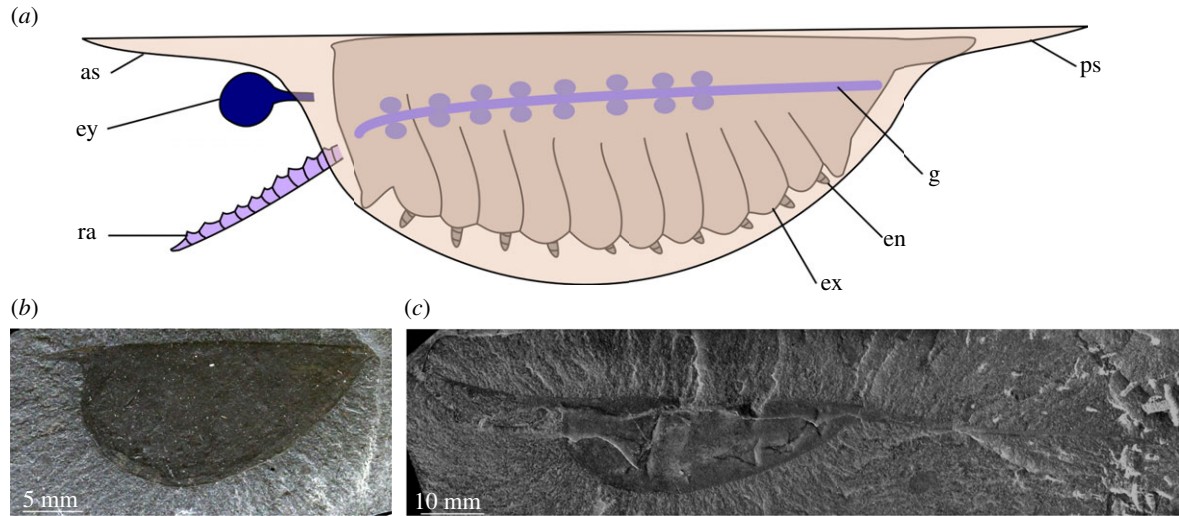

**Figure 2.** Morphology of *Isoxys*. (*a*) line drawing of idealized *Isoxys* illustrating known soft parts. (*b*) YPM IP 005804, *Isoxys acutangulus* from the Burgess Shale, British Columbia, Canada (Cambrian: Wuliuan) (credit: W. K. Sacco). (*c*) USNM PAL 189170, *Isoxys longissimus* from the Burgess Shale, British Columbia, Canada (Cambrian: Wuliuan). Image courtesy of the Smithsonian Institution (EZID: http://n2t.net/ark:/65665/m372f2a644-97c3-441c-87e2-24b1dccb2e8c, credit: Xingliang Zhang). Abbreviations: as, anterior spine; en, endopod; ex, flap like exopod; ey, eye; g, gut with paired diverticulae; ps, posterior spine; ra, raptorial appendage. (Online version in colour.)

## 2. Material and methods

### (a) Outline analyses

Two-dimensional outlines of 20 *Isoxys* species, its sister taxon *Surusicaris elegans* [22], 6 *Tuzoia* species and 11 *Gnathophausia* species were constructed in Inkscape from the literature sources by S.P. (electronic supplementary material, table S1) and imported into R [23] for elliptical Fourier analysis (EFA). D.A.L. independently constructed 20 *Isoxys* and 1 *Surusicaris* outlines directly from photographs of fossil specimens to allow assessment of the error introduced in the creation of outlines (electronic supplementary material, table S2; and S1). *Tuzoia* was chosen for comparison as it is also common in Cambrian communities, and has been suggested to be closely related to *Isoxys* based on similarities in the structure of the eyes and carapace shape (e.g. [24]). *Gnathophausia* was selected because similarities in the carapace morphology of one species (*G. zoea*) and *Isoxys* have been repeatedly noted (e.g. [16,19]), the carapaces of these animals are similar in size (10–30 mm), and multiple species of *Gnathophausia* are known to be vertically mobile in the modern ocean, having been sampled from surface waters and at depths of over 3000 m [25].

Outlines were sampled at the same resolution (64 points provided sufficient detail to distinguish taxa), centred, scaled by centroid size and subjected to EFA using the *Momocs* package [26]. Harmonics describing 99.9% of the variation were retained. EFA results were visualized with a principal components analysis. A hierarchical clustering analysis (cluster package; [23]) quantitatively grouped similarly shaped carapaces together using all principal components.

### (b) Computational fluid dynamics

*Isoxys* species reflecting the variation in carapace shape over both PC1 and PC2 were chosen for inclusion in CFD simulations, and *Gnathophausia zoea* was analysed for comparison. We chose to undertake a two-dimensional analysis to conserve computational resources and minimize errors in the modelled geometries (the exact three-dimensional shape is unknown for most taxa). This is justified because undeformed *Isoxys* specimens preserved in dorsal view show a narrow profile [16,21,27,28]. Propulsion during swimming derived from movement of the ventral appendages, and not from flapping of the bivalve carapace [15,27]. A lack of adductor muscles means that *Isoxys* was unable to alter the size of the gape during swimming [15], and the numerous specimens preserved in 'butterfly' orientation are considered exuviae [27]. In addition, the full variation of the carapaces considered in this study can be visualized in two dimensions. Two-dimensional analyses are commonly performed on analyses of wing outlines to assess aerodynamic performance by both biologists and engineers, (e.g. [29,30]). A two-dimensional analysis is suitable for this study as the wake behind the *Isoxys* carapaces is steady at the Reynolds numbers considered, so can be modelled with two-dimensional simulations [31,32]. The impact of soft parts such as eyes protruding from the carapace was analysed for two taxa: *Isoxys acutangulus* and *I. longissimus*. *Isoxys* was assumed to be negatively buoyant, like *Gnathophausia* [33]. All *Isoxys* species were assumed to have the same carapace composition and density. The cuticle ornamentation in some *Isoxys* species is not expected to impact the drag at the low Reynolds numbers considered in this study, as the roughened surface falls within the slowly moving fluid near the carapace surface [31].

Following validation and verification of the model and set-up for low Reynolds numbers, and mesh quality assessments using ANSYS Mesh and ANSYS Fluent (Ansys Academic, Release 2020 R2; electronic supplementary material, S2), outlines of the selected *Isoxys* species and *G. zoea* were exported from R as .txt files readable by ANSYS DesignModeler (Ansys Academic, Release 2020 R2; electronic supplementary material, S2), and standardized to a dorsal length (chord length) of 25 mm. This allowed size-independent comparisons of hydrodynamic performance of shapes. While some *Isoxys* taxa (e.g. *I. communis*, *I. longissimus*) can reach up to 50 mm, other adult forms only reach approximately 20 mm (e.g. *I. glaessneri*, *I. volucris*) [18,27]. A size of 25 mm represents a compromise size for comparison between these larger and smaller forms, with the influence of larger size able to be assessed by simulating higher *Re* (as *Re* depends on both size and swimming speed).

CFD simulations were conducted using the steady-state laminar solver in ANSYS Fluent (Ansys Academic, Release 2020 R2). The laminar solver performed best of the three considered during validation and verification (laminar, SST, k-epsilon; electronic supplementary material, S2), as expected for the low Reynolds numbers in this study [31].

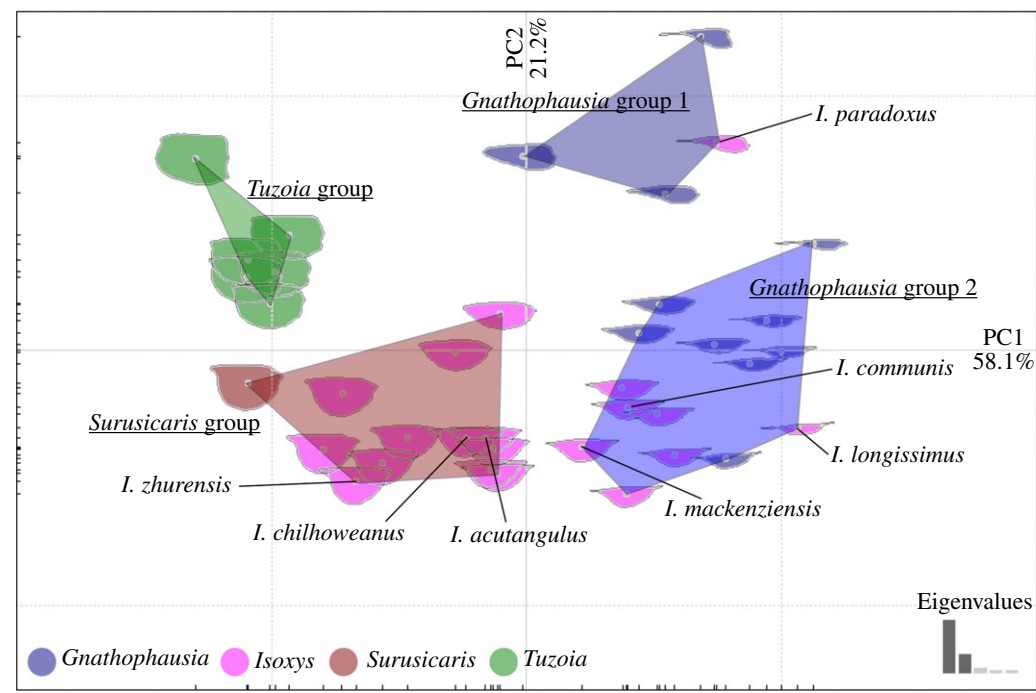

**Figure 3.** Principal component analysis of results of EFA conducted on the outlines of 11 species of *Gnathophausia*, 20 *Isoxys*, 1 *Surusicaris* and 6 *Tuzoia*. Convex hulls indicate optimum four groupings as recovered by clustering analysis. Labelled *Isoxys* species chosen for subsequent hydrodynamic analysis. (Online version in colour.)

Coefficients of drag (Cd) and lift (Cl) (electronic supplementary material, S2) were calculated under three flow speeds equating to 0.75, 1.00 and 1.18 body lengths per second (chord Reynolds numbers, $Re$, 255, 340 and 400 respectively for saltwater conditions at 0°C; electronic supplementary material, S2). These $Re$ were chosen as swimming speeds of between 75% and 100% of the body length per second have been observed in adult *Gnathophausia ingens* (carapace mean length approx. 25 mm [34]), and as the laminar model was validated against published drag and lift data for NACA aerofoils at exactly $Re = 400$ [29]. The chord length (25 mm) was taken as the reference area for both coefficients.

Solutions were considered converged when residuals were less than $10^{-6}$. Simulations were run at numerous angles of attack, to evaluate the hydrodynamic performance of carapaces at multiple orientations. In tank experiments, *Gnathophausia ingens* has been observed to change angle of attack to generate more lift or less drag at different swimming speeds [34]. The angle of attack was increased from 0 to 8° at all $Re$, until the stall angle could be identified and/or unsteady flow was observed. If the stall angle was not reached, further experiments were run until the maximum lift coefficient was obtained. Negative angles of attack were also simulated to assess the negative lift generated by the different outlines. In all cases, the absolute value of the negative angle of attack was increased until the drag coefficient was equal to or greater than that of the stall angle. When unsteady flow was suspected to be the reason that steady-state simulations did not converge, the steadiness of the flow field was determined by carrying out a time-dependent analysis of 100 time steps, with each time step equal to the flow speed (so, for an inlet velocity of 0.01875 ms$^{-1}$, the time step = 0.01875 s).

## 3. Results

### (a) Outline analysis

In the outline analysis, a total of 18 harmonics were retained. Principal coordinates 1 and 2 described 79.3% of the total variation. Carapace asymmetry, narrowness and spine length increased as PC1 became more positive, while the length of the anterior spine relative to the posterior spine corresponded to an increase in PC2 (figure 3). *Isoxys* occupied the largest area in the morphospace. Visual overlap of the areas occupied by the genera demonstrated that some *Isoxys* taxa were more similar in shape to *Gnathophausia* than their Cambrian relatives *Surusicaris* and *Tuzoia*. Confirmation was provided by the cluster analysis (figure 3). All six *Tuzoia* species formed a single cluster, with *Isoxys* taxa spread over the three remaining groups (*Surusicaris* group, *Gnathophausia* groups 1 and 2). Species clustered with *Surusicaris* have symmetric and deep carapaces and relatively short spines. Species in the *Gnathophausia* groups displayed narrower outlines whose narrowness, asymmetry and spine length increased with PC1. The single species of *Isoxys* in *Gnathophausia* group 1, *I. paradoxus*, displayed an anterior spine much more elongate than its posterior one, that contrasted with the seven species in *Gnathophausia* group 2 with their spines of approximately equal length.

### (b) Computational fluid dynamics

CFD simulations assessed the impact of increasing asymmetry, spine length and relative lengths of anterior and posterior spines on hydrodynamic performance. Inclusion of eyes did not significantly impact the hydrodynamic performance of carapaces (electronic supplementary material, S3). *Isoxys zhurensis*, the most symmetric species chosen for analysis, created an unsteady wake with Kármán vortex street at the lowest $Re$ considered, and so no drag or lift coefficients were obtained (electronic supplementary material video). The flow around remaining *Isoxys* outlines was laminar with a steady wake, and there was no evidence for three dimensionality (electronic supplementary material, S4). Greater asymmetry and narrowness (more positive in PC1, figure 3) resulted in lower drag coefficients, as demonstrated by a comparison of the short-spined taxa *I. chilhoweanus*,

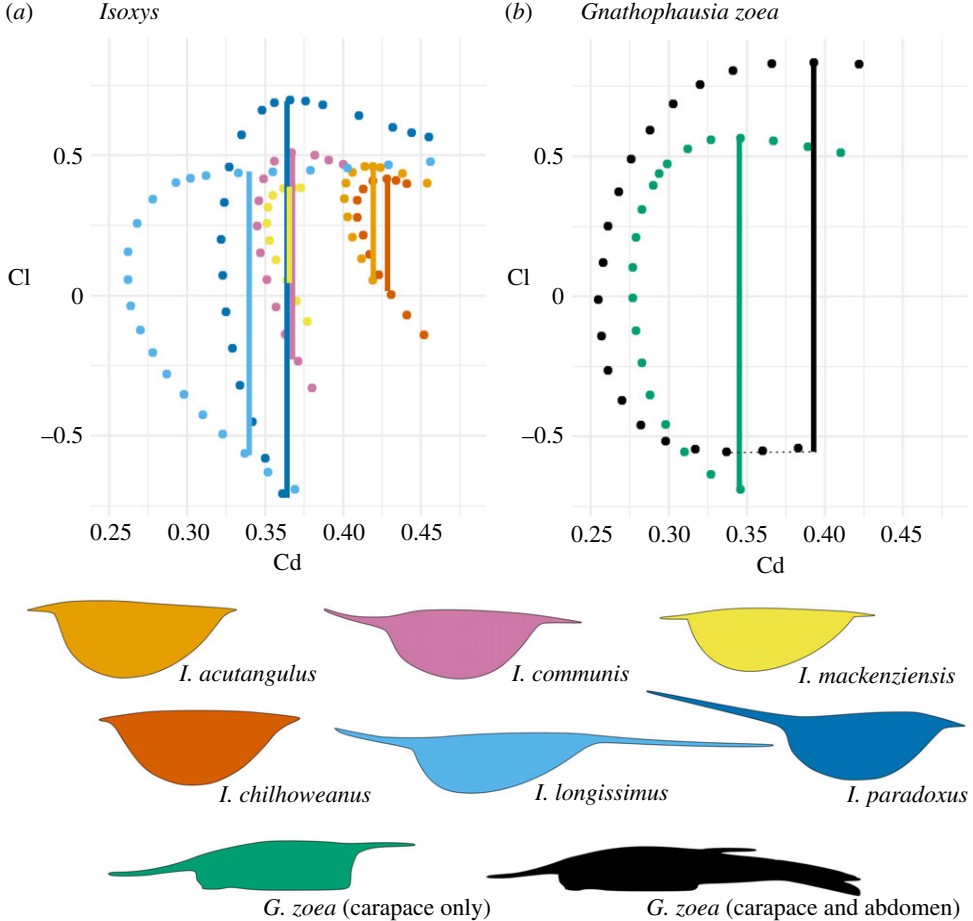

**Figure 4.** Drag polars (plot of Cd against Cl) of taxa analysed at $Re = 255$ (0.75 body lengths per second for an animal 25 mm long). Each point corresponds to a single simulation at a different angle of attack. Vertical bars show range of lift coefficients. Note that flow was unsteady for Isoxys zhurensis at $Re = 255$, and so no quantitative lift or drag coefficients were recorded. Drag polars at faster flow speeds and raw data presented in electronic supplementary material, S3. (Online version in colour.)

*I. acutangulus* and *I. mackenziensis.* The most asymmetric of these forms, *I. mackenziensis*, produced lower drag coefficients than the other two, but ranges of lift coefficients were similar for all three (figure 4a). More elongate spines increased the range of lift coefficients (vertical bars, figure 4a) and, significantly, negative lift coefficients at negative angles of attack (e.g. compare *I. mackenziensis*, *I. communis* and *I. longissimus*). In *I. paradoxus*, where the anterior spine is much longer than the posterior one (more positive in PC2, figure 3), the range of lift coefficients further increased (figure 4a). Similar drag coefficients and ranges of lift coefficients were obtained in an analysis of the hydrodynamics of *Gnathophausia zoea*, when either the carapace alone or both the carapace and abdomen were considered (figure 4b).

## 4. Discussion

### (a) Vertical migrations and niche partitioning in *Isoxys*

Functional morphology of *Isoxys* fossil specimens supports an off-bottom (hyperbenthic or pelagic) life habit for this animal [15,16,18,19,32], based on the eye orientation (forwards, slightly ventral) and the elongate slender carapace shape of *Isoxys*. Our study combines outline morphometric and CFD analyses and suggests that *Isoxys* species occupied a variety of niches, including some as pelagic vertically mobile predators.

Lift and drag coefficients of *Isoxys* carapaces indicate variation in the depth range and swimming speeds of these species. *Isoxys* taxa clustering with *Surusicaris* (figure 3)

generate positive lift, but do not generate negative lift (figure 4). This supports suggestions of previous workers that these *Isoxys* species may have occupied a hyperbenthic (1–10 m above the seafloor) and/or nektobenthic [21] niche, perhaps moving vertically short distances in the water column [15,19,35]. Vertical movement would be achieved by altering the angle of attack to produce lift force greater than (ascent), equal to (horizontal swimming), or less than (descent) the impact of their negative buoyancy. Drag reduction associated with streamlining would have allowed some taxa (e.g. *I. mackenziensis*) to capture faster-moving prey animals, as faster swimming speeds could have been maintained over longer distances for the same metabolic cost.

*Isoxys* species clustering with *Gnathophausia* show convergent adaptations to moving vertically in the water column, such as asymmetric carapaces with elongate anterior and posterior spines (figures 3 and 4). This does not preclude elongate spines from also acting as anti-predatory deterrents, as suggested by Vannier & Chen [15]. These adaptations provided hydrodynamic benefits that would have allowed the *Isoxys* species to operate over a wider bathymetric range. A streamlined carapace facilitates not only faster movement, but also more efficient swimming, beneficial for migrations over a long distance. The carapace shapes of *I. longissimus* and *I. paradoxus* generate lift coefficient ranges and minimum drag coefficients comparable to the modern crustacean *Gnathophausia zoea* (figure 4), which has been recovered at depths ranging from surface waters down to approximately 3000 m in the modern ocean [25]. These results also suggest

that an elongate abdomen in *G. zoea* reduces the drag experienced by the animal slightly but does not greatly impact on the range of lift coefficients (figure 4*b*), though the abdomen may also play a physical role. Animals that move vertically in the water column do not have to cover the entire distance from the surface ocean to demersal communities, and instead sometimes migrate across only a shorter vertical distance. Thus, *Isoxys* taxa with the broadest ranges of lift coefficients (*I. longissimus* and *I. paradoxus*) probably covered a wider depth range than those with smaller ranges of lift coefficients (e.g. *I. communis*).

Corroborating evidence for variation in bathymetric range for different *Isoxys* species comes from the fossil record itself. Members of different groups as resolved in the cluster analysis (convex hulls, figure 3) co-occur with different relative abundances in Cambrian deposits preserving soft-bodied fossils. In general, species with inferred vertically migrating lifestyles are much rarer than those that lived close to the seafloor. In the Chengjiang Biota, *Isoxys auritus* (*Surusicaris* group) greatly outnumbers both *I. paradoxus* and *I. curvirostratus* (*Gnathophausia* groups 1 and 2, respectively) (figure 3; [15,21,36–38]). A similar pattern of relative abundances can be observed in the two Burgess Shale taxa (figure 2): in the Walcott Quarry, *I. acutangulus* (*Surusicaris* group) comprises nearly 0.5% of the total community, vastly outnumbering the extremely rare inferred vertical migrant *I. longissimus* (*Gnathophausia* group 2 [16,39]). The relative abundances of these *Isoxys* species can be partly explained by the differences in lifestyle predicted by the carapace outline and soft anatomy. The less hydrodynamic taxa (those with higher drag coefficients and narrower ranges of lift coefficients) probably lived near the seafloor, with the more streamlined species living in the water column and occupying a broader bathymetric range. This broader bathymetric range would have included more open water settings, beyond the maximum depth of the shelf where Cambrian deposits preserving soft-bodied fossils occur—*Gnathophausia zoea* for example has been found at depths of up to 3000 m [25]. As modern euarthropod carapaces disarticulate quickly after death (e.g. [40]), the preservation potentials for pelagic euarthropods living high in the water column are lower than for those living closer to the seafloor. In addition, animals which occupy an ecological niche in the open water are less likely to find themselves over shelf environments such as those which preserve soft-bodied fossils or be trapped and transported by an obrution event responsible for the preservation of soft-bodied communities in these settings. The small numbers of vertically mobile *Isoxys* individuals observed may have been at the bottom of their vertical migrations and/or been transported horizontally by currents. *Isoxys* species are not globally distributed [41]. Many species (for example those clustering with *Surusicaris*) appear suited to hyperbenthic habits, and so would be expected to have provincial distributions. The limited geographical distribution of *I. longissimus* and *I. paradoxus* is most likely to be due to a combination of factors. Firstly, deposits where *Isoxys* is expected to be preserved are not evenly distributed in time and space—Stage 3 deposits are mostly in South China, while Wuliuan and younger are mostly in Laurentia [42], though the absence of the Chengjiang species *I. paradoxus* from Sirius Passet is notable. Secondly, the lower preservation potential of pelagic (compared to hyperbenthic) species means that they are rare even in Tier 1 Burgess

Shale-type Lagerstätten (*sensu* [42]). However, despite its rarity, the Burgess Shale species *I. longissimus* has a wider known geographical range than the co-occurring *I. acutangulus*. The former has also been reported from the Wheeler Formation, House Range, UT, USA [43]. The situation appears more complex in the Emu Bay Shale, where the more hydrodynamic species *I. communis* greatly outnumbers the less streamlined *I. glaessneri* [27]. However, the Emu Bay Shale is not a traditional Burgess Shale-type deposit, instead it represents a localized deep-water micro basin on the inner shelf [44]. Here fluctuating oxygen levels may have periodically deoxygenated the water column, possibly killing pelagic taxa like *I. communis* in great numbers and creating a taphonomic bias that preferentially preserves pelagic taxa.

Further support for the Chengjiang taxon *Isoxys auritus* occupying a niche closer to the seafloor than *I. curvirostratus* comes from a comparison of the soft anatomy (soft parts are unknown for *I. paradoxus*) [21]. The stout endopods of *I. auritus* appear well suited for interacting with the substrate, while exopods with broad fringing lamellae and a sophisticated vascular system in the more streamlined *I. curvirostratus* suggest it was a more powerful swimmer, providing additional support for a pelagic habit [21].

Lastly, a compendium of fossil, geochemical and phylogenetic data show that vertically mobile *Isoxys* species would have had access to a variety of pelagic prey items and an oxygenated water column to travel in. Cambrian oceans were not stratified, instead displaying wedge-shaped oxygen minimum zones broadly comparable to modern oceans [45]. *Isoxys* prey size range (approx. 5–20 mm [19]) includes Cambrian phytoplanktivores, such as crustaceans with setae and filter plates from Sirius Passet and Mount Cap (15–50 mm) [4,8,9,12], as well as crown-group branchiopods and copepods and total group ostracods [46], as well as bradoriids, some of which were also likely pelagic [47,48]. The presence of planktonic larvae, another possible prey item, can be inferred from tip-dated phylogenetic analyses that support the evolution of metamorphosis in euarthropods by the Cambrian [49]. Thus, a range of different data sources suggest multiple *Isoxys* taxa were vertically mobile, and that the Cambrian ocean could support such an ecology.

## (b) Metazoans and the Cambrian biological pump

The presence of the likely vertically mobile *Isoxys paradoxus* in the Chengjiang Biota makes it the oldest confidently identified euarthropod vertical migrant, and probably among the first animals to employ this life habit. For this vector to be significant by the Cambrian Stage 3, a large biomass of *Isoxys* would need to move vertically. While pelagic animals have a lower preservation potential than benthic ones (for example, very few fossil copepods are known [50]), *Isoxys* species with inferred (hyper)benthic habits are extremely abundant in both the Chengjiang and Burgess Shale [15,16,21,36–38]), and their pelagic counterparts *Isoxys longissimus* and *I. paradoxus* may have been similarly numerous. The Chengjiang and coeval Qingjiang biotas also preserve the earliest evidence for gelatinous zooplankton, which move vertically small distances in the modern ocean [51,52]. However, an active swimming *Isoxys* would have covered greater distances more rapidly. Furthermore, the presence of a through gut would have increased processing time for food, vital for transporting nutrients consumed in surface

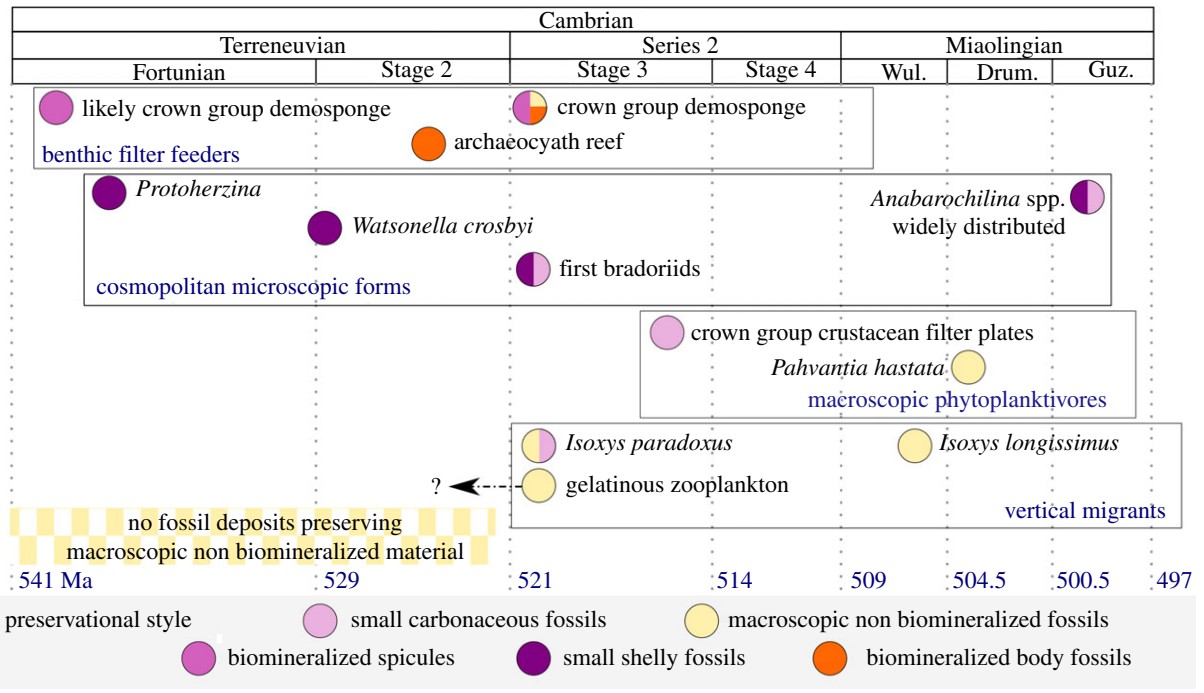

**Figure 5.** First known appearance in the fossil record of metazoans that impacted the biological pump. Circle colour denotes preservation style. Fossils with more than one preservation style indicated with split circles, with the preservation style that provided the oldest evidence on the left of the circle. Abbreviations: Drum., Drumian; Guz., Guzhangian; Wul., Wuliuan. (Online version in colour.)

waters to the deep ocean as faecal pellets. Vertical migration was one of many important eukaryotic and metazoan innovations key to establishing a modern-style biological pump. A series of metazoan innovations which appear in the fossil record in quick succession during the early Cambrian gave the biological pump a modern-looking structure (figure 5), which was strengthened during the Phanerozoic.

The shift to primary production dominated by eukaryotes, and the innovation of active suspension feeding, would have ventilated the oceans, cleared organic matter in the water column, and increased transfer of oxygen and plankton of increasing sizes from the surface to the sediment–water interface (e.g. [6]). The first of these events occurred during the Cryogenian [7], but while benthic passive suspension feeders are known from the Ediacaran (e.g. [53]), early Cambrian stem and crown-group sponges represent the oldest benthic active suspension feeders [10,11]. As active suspension feeders, sponges were able to transport large volumes of water between benthic and pelagic realms (e.g. [54]), and the most abundant sponges from the Cambrian, the reef building archaeocyaths, display pore size differentiation within reef systems [55], presumed to be evidence of prey size-selectivity. This illustrates that predator–prey feedbacks were present in the Cambrian Stage 2 (figure 2), presenting a potential driver towards a larger size of plankton, increasing the sinking speed and efficiency of this part of the biological pump, and ventilating the water column.

The invasion of the pelagic realm by eumetazoan zooplankton provides the next step towards a modern-style biological pump. These zooplankton would have further cleared surface waters and contributed faecal pellets to organic aggregates sinking to the deep ocean, and also increased oxygen levels at depth (e.g. [6,54]). The small shelly fossil (SSF) record provides a source of evidence for the invasion of the pelagic realm by eumetazoans. Terreneuvian SSFs include the possible chaetognath *Protoherzina* and the molluscs *Watsonella*, *Aldanella* and *Oelandiella*, whose widespread distributions are suggestive of a pelagic lifestyle, or at least a planktonic larval stage (figure 5) [56]. Euarthropods, likely early occupants of the plankton [35], are represented in the SSF record from the Cambrian Stage 3 by millimetre-scale bradoriids among others (figure 5; e.g. [47,57,58]). The first macroscopic nektonic suspension feeders, such as the radiodont *Tamisiocaris*, also appear at this time [59], while the first centimetre-scale phytoplanktivores are identified close to the Stage 3–4 boundary (figure 5) [12]. These data suggest that there was an increase in the diversity of millimetre-scale zooplankton at or close to the base of Stage 3, very close in time to the appearance of the first vertical migrants. Most bradoriids are considered benthic, however *Anabarochilina* increased its distribution in three phases, providing complementary evidence for a steady strengthening of the pump during the Cambrian. In Epoch 2 *Anabarochilina* was coupled with benthic assemblages, by the Wuliuan it spread to a wider spectrum of lithofacies, and by the Guzhangian two species became widely distributed [48].

## (c) Significance of vertical migration for the early Cambrian radiation of animals

The metazoan innovation of vertical migration would have impacted both demersal and pelagic communities. The strength of the impact depends on the amount of biomass undertaking vertical migration. Models based on Cambrian environmental parameters predict that vertical migration would have increased the efficiency of the carbon pump by around 7% [60], however, more significantly, vertical migrants transport organic nutrients to the deep sea more quickly than aggregates, with a different nutritional balance, and repackage decaying sinking organic matter (figure 1) [1,13,61–64]. In addition, vertical migrants are major contributors to ocean mixing and ventilation, spreading oxygen

and nutrients throughout the water column (e.g. [54,65]). These effects probably played a role in contributing to the rapid rate of diversification during the Cambrian explosion, interwoven with numerous evolutionary and ecological feedbacks. For example, the higher metabolic needs and nutrient requirements of large biomineralizing animals and motile predators [66] may have been facilitated by increased quality and quantity of nutrient transport in the biological pump, and resulted in increased oxygenation of bottom waters. In turn, the increasing size and motility of predators would have provided a further ecological pressure for animals to 'escape' into the pelagic realm.

The establishment of a biological pump with a modern-style architecture by the Cambrian Stage 3 does not mean that the fluxes along the aggregation and vertical migration vectors (figure 1) remained constant to the modern day. Indeed they likely strengthened through the Palaeozoic with an increase in biomass (from an increased number of taxa, individuals and size of individuals). Fossil evidence points to additional metazoan innovations during the Palaeozoic that would have affected the fluxes along these vectors and strengthened the pump. For example, the aggregation vector would have been strengthened following the evolution of centimetre-scale and decimetre-scale phytoplanktivores later in the Cambrian [12,67], and the major radiation of plankton and the evolution of metre-scale nektonic suspension feeders during the Great Ordovician Biodiversification Event [68–70]. The flux of nutrients along the vertical migration vector would have increased as pelagic and vertically migrating animals diversified and increased in size—an innovation that would also have increased the mixing of

waters and ocean ventilation. For example, the evolution of large, fast moving fish as part of the Devonian nekton revolution is expected to have been especially significant [54,71].

In summary, the innovation of vertical migration in some *Isoxys* species was one of several interwoven and coevolutionary feedbacks during the early Cambrian that increased the efficiency and altered the architecture of the biological pump, likely contributing to the rapid expansion in metazoan diversity at this time.

Data accessibility. All data and supplementary materials are available through the Open Science Framework: https://doi.org/10.17605/OSF.IO/2JDRS.

Authors' contributions. S.P.: conceptualization, data curation, formal analysis, funding acquisition, investigation, methodology, project administration, resources, software, validation, visualization, writing–original draft, writing-review & editing; A.C.D.: writing–review & editing; D.A.L.: data curation, writing–review and editing; I.A.R.: methodology, validation, writing–review & editing. All authors gave final approval for publication and agreed to be held accountable for the work performed therein.

Competing interests. We declare we have no competing interests.

Funding. S.P. was supported by an Alexander Agassiz Postdoctoral Fellowship (Harvard University) and a Herchel Smith Postdoctoral Fellowship (University of Cambridge), D.A.L. by a Dame Kathleen Ollerenshaw Research Fellowship (University of Manchester), and I.A.R. by a Museum Research Fellowship (Oxford University Museum of Natural History).

Acknowledgements. We thank the associate editor, two anonymous referees, and Christian Klug, who provided helpful reviewer comments. We thank members of the Ortega-Hernández Lab for Invertebrate Paleobiology (Harvard University) for fruitful discussions. S. Butts (Yale Peabody Museum) and M. Florence (Smithsonian National Museum of Natural History) provided curatorial assistance.

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
