## [Peer Review File · Proceedings of the Royal Society B: Biological Sciences]

Review History

RSPB-2021-0464.R0 (Original submission)

Review form: Reviewer 1 (Vincent Perrier)

Recommendation

Accept with minor revision (please list in comments)

Scientific importance: Is the manuscript an original and important contribution to its field?

Excellent

General interest: Is the paper of sufficient general interest?

Excellent

Quality of the paper: Is the overall quality of the paper suitable?

Excellent

Is the length of the paper justified?

Yes

Should the paper be seen by a specialist statistical reviewer?

No

Do you have any concerns about statistical analyses in this paper? If so, please specify them explicitly in your report.

No

It is a condition of publication that authors make their supporting data, code and materials available - either as supplementary material or hosted in an external repository. Please rate, if applicable, the supporting data on the following criteria.

Is it accessible?

Yes

Is it clear?

Yes

Is it adequate?

Yes

Do you have any ethical concerns with this paper?

No

Comments to the Author

This paper represents a very valuable contribution to our knowledge of the origin and evolution of the pelagic communities in the Early Palaeozoic and it provides for the first time data suggesting the presence of vertically migrating zooplankton in Early Cambrian oceans. The paper is well written and figures/supplementary information are of very good quality and thus the paper could be published as it is.

I just have a few questions/suggestions that may or may not enter the scope of this paper and that motivated my recommendation to "accept with minor revisions":

- Recent planktonic organism (including the *Gnathophausia* species) and other Palaeozoic supposedly planktonic taxa (e.g. graptolites, conodonts, myodocope ostracods...), have a cosmopolitan/transoceanic distribution. How can you explain that it is not the case for (at least some) *Isoxys* species? Is that related to taphonomic processes and/or the lack of Lagerstätten in which they could be preserved? A palaeobiogeographic distribution map of the mentioned *Isoxys* species and a recent distribution map of *Gnathophausia* species would be valuable additions to the paper.

- Nowhere in the text is mentioned the possible role of the cuticle ornamentation in the hydrodynamics of the carapace. Several species of *Isoxys* and *Tuzoia*, as well as recent planktonic crustaceans, have micro-ornament (e.g. reticulations, corrugations) on their carapace. Is it possible that it played a role in the hydrodynamics of the animal or is it a minor factor compared to the overall shape of the carapace?

- Could there be an impact of the position of the swimming appendages on the hydrodynamics of the animal? *Gnathophausia* have its appendages outside its cephalic shield (a bit like a propeller pushing the streamlined front of the animal). *Isoxys*, on the other hand, has its appendages inside/below the bivalve carapace, wouldn't that be a problem for vertical migration?

- Maybe an interesting additional reference regarding putative planktonic Bradoriids: Williams, M., Vandenbroucke, T. R., Perrier, V., Siveter, D. J., & Servais, T. (2015). A link in the chain of the Cambrian zooplankton: bradoriid arthropods invade the water column. *Geological Magazine*, 152(5), 923-934.

Review form: Reviewer 2

Recommendation

Reject – article is scientifically unsound

Scientific importance: Is the manuscript an original and important contribution to its field?
Acceptable

General interest: Is the paper of sufficient general interest?
Acceptable

Quality of the paper: Is the overall quality of the paper suitable?
Poor

Is the length of the paper justified?
Yes

Should the paper be seen by a specialist statistical reviewer?
No

Do you have any concerns about statistical analyses in this paper? If so, please specify them explicitly in your report.
No

It is a condition of publication that authors make their supporting data, code and materials available - either as supplementary material or hosted in an external repository. Please rate, if applicable, the supporting data on the following criteria.

Is it accessible?
Yes

Is it clear?
Yes

Is it adequate?
Yes

Do you have any ethical concerns with this paper?
No

Comments to the Author

By employing Morphometric analyses and by modelling the hydrodynamic behaviors of the carapaces of various Cambrian arthropods *Isoxys* spp., Pates et al. explore the multiple morphospaces that the different species of *Isoxys* occupied, and discuss their capability/probability of migrating vertically in the Cambrian ocean. They argue that these bivalved arthropods conducted active migration in the water column. They concluded that by doing this *Isoxys* and the related/similar arthropods could have connected the oceanic sub-ecosystems at different fathoms, which is analogous to the so-called "biological pump" described from modern ocean.

This study is innovative, not only in the methods it utilizes, but as well in the perspective it explores the palaeo-ocean. And the topics it discusses, i.e., the role of a widely distributed arthropod group in the early – middle Cambrian oceanic ecosystem, and the structure and function of such early ecosystem on Earth in comparison to that of today, is potentially of scientific importance. However, I cannot recommend publication of this manuscript in Royal Society, for the following major concerns:

1) The presence of biological pump in the Cambrian period is based on weak, limited evidence. The authors conducted hydrodynamic simulation on only the carapace. The swimming capacity, however, does not only depend on its carapace design, but probably more importantly on

appendage organization. The latter has been totally ignored. Also, I'm not sure about the credit of the hydrodynamic modulation. For one thing, it is difficult to know the exact width of the carapace due to significant compression in the Burgess-type fossils. This would significantly bias the analysis. Meanwhile, the three-dimensional shapes of the animals' body cannot be determined; probably they are not fixed: the animals could have adjusted the angle between two valves during swimming to obtain different hydrodynamic performances. Please imagine extreme occasions where *Isoxys* swims in a "butterfly" position, or with their valves only slightly open. The lift, drag forces would be completely different. In addition, although the authors claimed that the effect of the eyes can be ignored, this is incredible given the sizes of the spherical eyes of *Isoxys*. The great appendages in a number of *Isoxys* species would have important effects on swimming too. All in all, it is not acceptable to base the hydrodynamic result on the compressed (deformed) carapace alone.

As suggestions, I would like the authors to include at least some soft-part data in their analyses, for example the eyes and the great appendages that significantly protrude beyond the carapace. I know this would be difficult, but the authors can choose to focus on a few species and perform more comprehensive analysis (e.g., considering different compression ratios, and different angles between valves while swimming, etc), rather than dabble many species.

2) If the authors want to argue about the vertical migration of *Isoxys*, they should not only explore the hydraulic design of the carapace and the propulsion of the appendages, but should also seek for fossil evidence that these animals are preserved/living at different environmental settings, in particular different depths of the water column.

3) The authors hypothesized that *Isoxys* species (and other similar bivalved forms) occupied different niches. This is based mainly on the morphometric analysis. However, shape variation occurs in many animal groups and can be the results of various adaptations. Therefore, shape variation can serve as only supplementary evidence for niche diversification. Again, I think soft anatomy would be (more) important. There are rich literatures describing the morphologies of the eyes and the great appendages, both of which are crucial for feeding and movement of *Isoxys*.

4) The discussions in section "Metazoans and the Cambrian biological pump" are generally good, but not well organized. I'm happy to see SSF is considered, but in general this section seems to have deviated from the main points of the manuscript, and have been based on poor evidence – too ambitiously covered many taxa, Lagerstätten and ecological concepts.

5) From the carapace design and known appendage structures of *Isoxys*, I believe it can swim. But I feel a bit reluctant to think that it can migrate across great depths. There are many other arthropods in Cambrian Lagerstätten that, judged from their body design, seem to be more adaptive to swimming, such as *Tokummia* and *Waptia*. As to me it is more favorable to explore the migrating capability of these animals.

Other comments are detailed in the annotated pdf. I hope these could help.

I suggest the authors to focus on only a few taxa, and to base the ecological conclusion on both carapace hydrodynamics and feeding and locomotory merits of the eyes and the frontal appendages of *Isoxys*, thus making the analyses more in-depth and convincing. It is also important to seek for fossil evidence for the presence of *Isoxys* in different depths of the Cambrian ocean. I agree that *Isoxys* and *Tuzoia* are important bivalved arthropods of the Cambrian, and should have played some role. But they may not be the best organisms for testing the biological pump hypothesis. Moreover as mentioned above, the present evidence is too weak for vertical migration.

Review form: Reviewer 3 (Christian Klug)

Recommendation

Accept with minor revision (please list in comments)

Scientific importance: Is the manuscript an original and important contribution to its field?
Excellent

General interest: Is the paper of sufficient general interest?
Excellent

Quality of the paper: Is the overall quality of the paper suitable?
Good

Is the length of the paper justified?
Yes

Should the paper be seen by a specialist statistical reviewer?
No

Do you have any concerns about statistical analyses in this paper? If so, please specify them explicitly in your report.
No

It is a condition of publication that authors make their supporting data, code and materials available - either as supplementary material or hosted in an external repository. Please rate, if applicable, the supporting data on the following criteria.

Is it accessible?
N/A

Is it clear?
N/A

Is it adequate?
N/A

Do you have any ethical concerns with this paper?
No

Comments to the Author

Dear Stephen, Allie, David and Imran,

this is a very nice and interesting paper, congratulations!

The paper is well written and a suitable contribution for the journal.

Here are a few things, which I think you can improve easily:

1. The title suggests that Isoxys alone made the first biological pump. I know, Isoxys is not super-rare and all, but it is also not so overwhelmingly abundant that you would expect it to revolutionize all the world's oceans of the time. I would formulate it a bit more humbly.
2. You discuss this later, but for me, to make a point that Isoxys was the first engineer of the biological pump, I would expect some evidence that they had a significant biomass that was migrating vertically. I would at least shortly discuss this in the methods, even if it is mentioning that they are not so super-common but that you think that this is due to a taphonomic bias, making the preservation of pelagic plankton less likely (with references).
3. You start the discussion with mentioning that Isoxys was not benthic. Please shortly state WHY people think this is the case.
4. I recommend to use 'migrant' uniformly instead of switching between migrator and migrant. Migrator sounds strange in my ears.
5. I would point out that the biological pump probably started out weakly and became stronger through the Phanerozoic. In a way, you do, but it gets a bit lost in details. Maybe you could try structuring the discussion a bit better.
6. I think the transport of oxygen is equally important to the nutrients, but oxygen is rarely

mentioned. Maybe stress this a bit more. See Butterfield's paper on the pelagic bioturbation. 7. The resemblance to thylacocephalans and phyllocarids is remarkable. Maybe it would be worthwhile discussing similarities shortly and mentioning that this kind morphology evolved convergently several times? Such an ecomorphological comparison might also help to strengthen some of your points.

I made some more remarks in the annotated pdf.

I look forward seeing this paper published!

Best wishes,

Christian

Decision letter (RSPB-2021-0464.R0)

04-May-2021

Dear Dr Pates:

Your manuscript has now been peer reviewed and the reviews have been assessed by an Associate Editor. The reviewers' comments (not including confidential comments to the Editor) and the comments from the Associate Editor are included at the end of this email for your reference. As you will see, the reviewers and the Editors have raised some concerns with your manuscript and we would like to invite you to revise your manuscript to address them.

Research ethics:

Use of animals and field studies:

It is a condition of publication that you make available the data and research materials supporting the results in the article. Please see our Data Sharing Policies (<https://royalsociety.org/journals/authors/author-guidelines/#data>). Datasets should be deposited in an appropriate publicly available repository and details of the associated accession number, link or DOI to the datasets must be included in the Data Accessibility section of the article (<https://royalsociety.org/journals/ethics-policies/data-sharing-mining/>). Reference(s) to datasets should also be included in the reference list of the article with DOIs (where available).

Please submit a copy of your revised paper within three weeks. If we do not hear from you within this time your manuscript will be rejected. If you are unable to meet this deadline please let us know as soon as possible, as we may be able to grant a short extension.

Best wishes,
Dr John Hutchinson
<mailto:proceedingsb@royalsociety.org>

Associate Editor

Board Member: 1

Comments to Author:

The authors report on the swimming ability of the Cambrian arthropod *Isoxys*. They ambitiously argue that their swimming ability was sufficiently strong that they connected oceanic sub-systems at different depths, acting as a "biological pump." This paper was reviewed by three expert reviewers along with myself. We all appreciated the ambitious aims of the paper and the attempt at the use of hydrodynamic methods to prove an interesting biological hypothesis.

However, we have several concerns with this paper that would prevent it from being accepted by the journal readership in its current form. Reviewer 1 and 2 both bring up similar concerns, although Reviewer 2 sees them as more critical. Namely, the reviewers suggest that distribution and biological pump ability of *Isoxys* should be compared to current planktonic taxa. Both reviewers also have concern with the accuracy of the hydrodynamic model. While I appreciate the elegance of a 2D model (see work by Jane Wang at Cornell that does 2-D models), justification of a 2D model is needed. 2D animals do not exist in nature, and results from a 2D simulation are only partially right. The authors may look into papers like: Effect of three-dimensionality on the lift and drag of nominal two-dimensional cylinders that discusses difference 2D and 3D approximations. Reviewers suggest that shell ornamentation, eyes, and appendages may play a role in drag coefficients, and some evidence should be given that these features can be neglected.

Reviewer 2 also worries that the compression of the shell in fossil record may influence the drag calculations. The authors do make some comparisons to drag coefficients of species *G. ingens*, which is appreciated. However, more evidence is needed for the role of the third dimension, perhaps with some test cases with three dimensional simulations. They are using 3D simulation software after all and not theory which generally favors 2D. Overall, Reviewer 2 need more evidence that *Isoxys* is a strong enough swimmer to traverse large distances. The authors give lift and drag coefficients, but the swimming speed and migration ability needs to be made more clear. More hydrodynamic justification and more evidence from extant species is needed to prove this authors' hypothesis. Reviewer 3 also suggests reduction of the claims in the title, in particular that *Isoxys* was the first biological pump. They also request consideration of the influence of oxygen. Please take into consideration comments of all reviewers in your revision.

Reviewer(s)' Comments to Author:

Referee: 1

Comments to the Author(s)

This paper represents a very valuable contribution to our knowledge of the origin and evolution of the pelagic communities in the Early Palaeozoic and it provides for the first time data suggesting the presence of vertically migrating zooplankton in Early Cambrian oceans. The paper is well written and figures/supplementary information are of very good quality and thus the paper could be published as it is.

I just have a few questions/suggestions that may or may not enter the scope of this paper and that motivated my recommendation to "accept with minor revisions":

- Recent planktonic organism (including the *Gnathopausia* species) and other Palaeozoic supposedly planktonic taxa (e.g. graptolites, conodonts, myodocope ostracods...), have a cosmopolitan/transoceanic distribution. How can you explain that it is not the case for (at least some) *Isoxys* species? Is that related to taphonomic processes and/or the lack of Lagerstätten in which they could be preserved? A palaeobiogeographic distribution map of the mentioned *Isoxys* species and a recent distribution map of *Gnathopausia* species would be valuable additions to the paper.

- Nowhere in the text is mentioned the possible role of the cuticle ornamentation in the hydrodynamics of the carapace. Several species of *Isoxys* and *Tuzoia*, as well as recent planktonic crustaceans, have micro-ornament (e.g. reticulations, corrugations) on their carapace. Is it possible that it played a role in the hydrodynamics of the animal or is it a minor factor compared to the overall shape of the carapace?

- Could there be an impact of the position of the swimming appendages on the hydrodynamics of the animal? Gnathophausia have its appendages outside its cephalic shield (a bit like a propeller pushing the streamlined front of the animal). Isoxys, on the other hand, has its appendages inside/below the bivalve carapace, wouldn't that be a problem for vertical migration?
- Maybe an interesting additional reference regarding putative planktonic Bradoriids: Williams, M., Vandenbroucke, T. R., Perrier, V., Siveter, D. J., & Servais, T. (2015). A link in the chain of the Cambrian zooplankton: bradoriid arthropods invade the water column. *Geological Magazine*, 152(5), 923-934.

Referee: 2

Comments to the Author(s)

By employing Morphometric analyses and by modelling the hydrodynamic behaviors of the carapaces of various Cambrian arthropods *Isoxys* spp., Pates et al. explore the multiple morphospaces that the different species of *Isoxys* occupied, and discuss their capability/probability of migrating vertically in the Cambrian ocean. They argue that these bivalved arthropods conducted active migration in the water column. They concluded that by doing this *Isoxys* and the related/similar arthropods could have connected the oceanic sub-ecosystems at different fathoms, which is analogous to the so-called "biological pump" described from modern ocean.

This study is innovative, not only in the methods it utilizes, but as well in the perspective it explores the palaeo-ocean. And the topics it discusses, i.e., the role of a widely distributed arthropod group in the early – middle Cambrian oceanic ecosystem, and the structure and function of such early ecosystem on Earth in comparison to that of today, is potentially of scientific importance. However, I cannot recommend publication of this manuscript in Royal Society, for the following major concerns:

1) The presence of biological pump in the Cambrian period is based on weak, limited evidence. The authors conducted hydrodynamic simulation on only the carapace. The swimming capacity, however, does not only depend on its carapace design, but probably more importantly on appendage organization. The latter has been totally ignored. Also, I'm not sure about the credit of the hydrodynamic modulation. For one thing, it is difficult to know the exact width of the carapace due to significant compression in the Burgess-type fossils. This would significantly bias the analysis. Meanwhile, the three-dimensional shapes of the animals' body cannot be determined; probably they are not fixed: the animals could have adjusted the angle between two valves during swimming to obtain different hydrodynamic performances. Please imagine extreme occasions where *Isoxys* swims in a "butterfly" position, or with their valves only slightly open. The lift, drag forces would be completely different. In addition, although the authors claimed that the effect of the eyes can be ignored, this is incredible given the sizes of the spherical eyes of *Isoxys*. The great appendages in a number of *Isoxys* species would have important effects on swimming too. All in all, it is not acceptable to base the hydrodynamic result on the compressed (deformed) carapace alone.

As suggestions, I would like the authors to include at least some soft-part data in their analyses, for example the eyes and the great appendages that significantly protrude beyond the carapace. I know this would be difficult, but the authors can choose to focus on a few species and perform more comprehensive analysis (e.g., considering different compression ratios, and different angles between valves while swimming, etc), rather than dabble many species.

2) If the authors want to argue about the vertical migration of *Isoxys*, they should not only explore the hydraulic design of the carapace and the propulsion of the appendages, but should also seek for fossil evidence that these animals are preserved/living at different environmental settings, in particular different depths of the water column.

3) The authors hypothesized that *Isoxys* species (and other similar bivalved forms) occupied different niches. This is based mainly on the morphometric analysis. However, shape variation occurs in many animal groups and can be the results of various adaptations. Therefore, shape variation can serve as only supplementary evidence for niche diversification. Again, I think soft

anatomy would be (more) important. There are rich literatures describing the morphologies of the eyes and the great appendages, both of which are crucial for feeding and movement of *Isoxys*.

4) The discussions in section “Metazoans and the Cambrian biological pump” are generally good, but not well organized. I’m happy to see SSF is considered, but in general this section seems to have deviated from the main points of the manuscript, and have been based on poor evidence – too ambitiously covered many taxa, Lagerstätten and ecological concepts.

5) From the carapace design and known appendage structures of *Isoxys*, I believe it can swim. But I feel a bit reluctant to think that it can migrate across great depths. There are many other arthropods in Cambrian Lagerstätten that, judged from their body design, seem to be more adaptive to swimming, such as *Tokummia* and *Waptia*. As to me it is more favorable to explore the migrating capability of these animals.

Other comments are detailed in the annotated pdf. I hope these could help.

I suggest the authors to focus on only a few taxa, and to base the ecological conclusion on both carapace hydrodynamics and feeding and locomotory merits of the eyes and the frontal appendages of *Isoxys*, thus making the analyses more in-depth and convincing. It is also important to seek for fossil evidence for the presence of *Isoxys* in different depths of the Cambrian ocean. I agree that *Isoxys* and *Tuzoia* are important bivalved arthropods of the Cambrian, and should have played some role. But they may not be the best organisms for testing the biological pump hypothesis. Moreover as mentioned above, the present evidence is too weak for vertical migration.

Referee: 3

Comments to the Author(s)

Dear Stephen, Allie, David and Imran,

this is a very nice and interesting paper, congratulations!

The paper is well written and a suitable contribution for the journal.

Here are a few things, which I think you can improve easily:

1. The title suggests that *Isoxys* alone made the first biological pump. I know, *Isoxys* is not super-rare and all, but it is also not so overwhelmingly abundant that you would expect it to revolutionize all the world's oceans of the time. I would formulate it a bit more humbly.
2. You discuss this later, but for me, to make a point that *Isoxys* was the first engineer of the biological pump, I would expect some evidence that they had a significant biomass that was migrating vertically. I would at least shortly discuss this in the methods, even if it is mentioning that they are not so super-common but that you think that this is due to a taphonomic bias, making the preservation of pelagic plankton less likely (with references).
3. You start the discussion with mentioning that *Isoxys* was not benthic. Please shortly state WHY people think this is the case.
4. I recommend to use 'migrant' uniformly instead of switching between migrator and migrant. Migrator sounds strange in my ears.
5. I would point out that the biological pump probably started out weakly and became stronger through the Phanerozoic. In a way, you do, but it gets a bit lost in details. Maybe you could try structuring the discussion a bit better.
6. I think the transport of oxygen is equally important to the nutrients, but oxygen is rarely mentioned. Maybe stress this a bit more. See Butterfield's paper on the pelagic bioturbation.
7. The resemblance to thylacocephalans and phyllocarids is remarkable. Maybe it would be worthwhile discussing similarities shortly and mentioning that this kind morphology evolved convergently several times? Such an ecomorphological comparison might also help to strengthen some of your points.

I made some more remarks in the annotated pdf.

I look forward seeing this paper published!

Best wishes,

Christian

Author's Response to Decision Letter for (RSPB-2021-0464.R0)

See Appendix A.

RSPB-2021-0464.R1 (Revision)

Review form: Reviewer 1 (Christian Klug)

Recommendation

Accept as is

Scientific importance: Is the manuscript an original and important contribution to its field?

Good

General interest: Is the paper of sufficient general interest?

Good

Quality of the paper: Is the overall quality of the paper suitable?

Good

Is the length of the paper justified?

Yes

Should the paper be seen by a specialist statistical reviewer?

No

Do you have any concerns about statistical analyses in this paper? If so, please specify them explicitly in your report.

No

It is a condition of publication that authors make their supporting data, code and materials available - either as supplementary material or hosted in an external repository. Please rate, if applicable, the supporting data on the following criteria.

Is it accessible?

Yes

Is it clear?

Yes

Is it adequate?

Yes

Do you have any ethical concerns with this paper?

No

Comments to the Author

Hi,

looks quite good now!

I suggest one final change (maybe at proofing stage:

line 303: Replace 'These zooplankton' by 'These planktic animals'

I look forward to its publication!
 Best wishes,
 Christian

Review form: Reviewer 2 (Vincent Perrier)

Recommendation

Accept as is

Scientific importance: Is the manuscript an original and important contribution to its field?

Excellent

General interest: Is the paper of sufficient general interest?

Excellent

Quality of the paper: Is the overall quality of the paper suitable?

Excellent

Is the length of the paper justified?

Yes

Should the paper be seen by a specialist statistical reviewer?

No

Do you have any concerns about statistical analyses in this paper? If so, please specify them explicitly in your report.

No

It is a condition of publication that authors make their supporting data, code and materials available - either as supplementary material or hosted in an external repository. Please rate, if applicable, the supporting data on the following criteria.

Is it accessible?

Yes

Is it clear?

Yes

Is it adequate?

Yes

Do you have any ethical concerns with this paper?

No

Comments to the Author

Dear Stephen, Allison, David and Imran,
 Congratulations for this nice paper that will be a very valuable contribution to our knowledge on arthropod zooplankton and on the origin and functioning of pelagic ecosystems. All my comments were addressed and taken into account in the revised version, so I do not have anything else to add except that I am looking forward to see it published!
 Congratulations again and best wishes to you all,
 Vincent

Decision letter (RSPB-2021-0464.R1)

03-Jun-2021

Dear Dr Pates

I am pleased to inform you that your manuscript entitled "Vertically migrating *Isoxys* and the early Cambrian biological pump" has been accepted for publication in Proceedings B. Congratulations!!

Data Accessibility section

Open Access

You are invited to opt for Open Access, making your freely available to all as soon as it is ready for publication under a CCBY licence. Our article processing charge for Open Access is £1700. Corresponding authors from member institutions (<http://royalsocietypublishing.org/site/librarians/allmembers.xhtml>) receive a 25% discount to these charges. For more information please visit <http://royalsocietypublishing.org/open-access>.

Paper charges

Sincerely,

Dr John Hutchinson

Associate Editor:

Board Member: 1

Comments to Author:

Congratulations. I look forward to seeing this work in print.

Appendix A

Editorial and referee comments and response

Original letter from the Editor in normal type, replies in **bold** below.

Associate Editor

Board Member: 1

Comments to Author:

The authors report on the swimming ability of the Cambrian arthropod *Isoxys*. They ambitiously argue that their swimming ability was sufficiently strong that they connected oceanic sub-systems at different depths, acting as a "biological pump." This paper was reviewed by three expert reviewers along with myself. We all appreciated the ambitious aims of the paper and the attempt at the use of hydrodynamic methods to prove an interesting biological hypothesis. However, we have several concerns with this paper that would prevent it from being accepted by the journal readership in its current form.

We appreciate the work of the Associate Editor and three referees to improve the manuscript. We have taken all the issues raised very seriously, and made numerous edits to our manuscript. We have also run a further 67 fluid dynamics simulations. We provide detailed responses to all of the suggestions below.

Reviewer 1 and 2 both bring up similar concerns, although Reviewer 2 sees them as more critical. Namely, the reviewers suggest that distribution and biological pump ability of *Isoxys* should be compared to current planktonic taxa. Both reviewers also have concern with the accuracy of the hydrodynamic model. While I appreciate the elegance of a 2D model (see work by Jane Wang at Cornell that does 2-D models), justification of a 2D model is needed. 2D animals do not exist in nature, and results from a 2D simulation are only partially right. The authors may look into papers like: Effect of three-dimensionality on the lift and drag of nominal two-dimensional cylinders that discusses difference 2D and 3D approximations.

While *Isoxys* was a three-dimensional animal, fossil specimens are preserved as compressions, in two dimensions and most commonly in lateral view (though a reasonable number are also known compressed obliquely and dorso-ventrally). We carried out two-dimensional simulations as: i) this approach minimizes errors in reconstructing modelled geometries; ii) *Isoxys* had a narrow profile, and so a two-dimensional simulation is equivalent to a plane through a three-dimensional simulation; and iii) it reduces computational demands. Furthermore, potential issues around two-dimensional simulations overpredicting drag and lift coefficients do not apply to the low Reynolds numbers at play, as the flow around *Isoxys* outlines created a steady wake.

We thank the Associate Editor for pointing us to the paper *Effect of three-dimensionality on the lift and drag of nominal two-dimensional cylinders* (Mittal & Balachandar 1995) as this provides a good overview of how flow develops with increasing Reynolds number around a cylinder, and helps to demonstrate our point that flow, in our study, can be accurately modelled using two-dimensional simulations as the flow created a steady wake. Vogel's *Life in Moving Fluids* also provides an overview of these transitions (pages 93–94 in paperback 2nd edition).

For a circular cylinder, at Reynolds numbers below 49 the wake is steady. At Re between 49 and 180 the wake is unsteady and the Kármán vortex street is observed. The wake becomes three dimensional at Re>180, and at this point a two-dimensional simulation would fail to capture the full effect of the three-dimensional flow field (Mittal & Balachandar 1995). It is this failure to

capture the three-dimensional effects leads to two dimensional simulations overpredicting the drag coefficient and peak-to-valley lift coefficients for flow regimes where the actual flow is three dimensional (Mittal & Balachandar 1995). This is because the suction pressure on the wake side of the cylinder drops when the flow becomes three-dimensional, as stresses in the near wake decrease due to the three-dimensionality of the flow (Mittal & Balachandar 1995) – a feature that is not picked up by two-dimensional simulations. Importantly for our study, the flow created a steady wake, and so we can be confident that the flow is not three-dimensional.

As *Isoxys* carapaces are more streamlined than the cylinder considered by Mittal & Balachandar (1995), the Reynolds numbers where these transitions occur will be higher (shapes closer to a bluff body will transition to unsteady flow and then 3D wakes at lower Re than more streamlined shapes). As all *Isoxys* taxa are much more streamlined than a cylinder, this transition to a three-dimensional wake would be expected to occur at $Re \gg 180$.

This expectation is confirmed in our study, as the wake is steady at all Reynolds numbers considered ($Re = 255, 340, 400$) in all cases (the one exception *Isoxys zhurensis* has an outline closest to a bluff body of all the taxa considered and is not included in the quantitative comparison). A steady-state laminar solver was used to gather the data presented. When the wake became unsteady and the Kármán vortex street was observed (because of increasing angle of attack and/or Reynolds number), we stopped collecting data. As the wake becomes unsteady at a lower Re than a 3D wake, by stopping collecting data when the wake became unsteady we can be sure that a 2D simulation is appropriate for the data we present.

We are confident that, for the Reynolds numbers and outlines simulated, a two-dimensional simulation is appropriate, and have demonstrated this with a three-dimensional example (Supplementary Materials 4).

Reviewers suggest that shell ornamentation, eyes, and appendages may play a role in drag coefficients, and some evidence should be given that these features can be neglected. Reviewer 2 also worries that the compression of the shell in fossil record may influence the drag calculations.

We thank the reviewers for bringing up these points, which are now dealt with more explicitly in the manuscript (see our responses to the Referees below). However, we do not expect ornamentation, eyes, or appendages to have a significant effect – the streamlining of the carapace will dominate under these flow conditions.

Shell ornamentation would serve to roughen the surface, which in specific flow conditions can reduce drag. However, for low Reynolds number laminar flow, as in this study, a roughened surface will not have a significant effect (Vogel, *Life in Moving Fluids* pages 99-101 in paperback 2nd edition). Further details are provided in the response to Referee 1 below.

We did not include eyes and appendages in our original analyses as we wanted to test features we considered to be under hydrodynamic control – namely the carapace shape – and not features that were under different selective pressures – namely the eyes and appendages. This allowed us to link carapace morphology to inferred ecology directly.

However, to demonstrate that the eyes did not dramatically reduce the hydrodynamic performance of *Isoxys* carapaces and to ensure that our comparison with *Gnathophausia* is appropriate, we conducted a total of 66 additional simulations on two new outlines of *Isoxys acutangulus* and *I. longissimus* that include eyes in two possible positions. These results are presented in Supplementary Materials 3, and demonstrate that the eyes did not have a significant

impact on the lift and drag coefficients of individual carapaces. This is most likely because they do not extend into the fast moving flow – the flow immediately around the carapace is slow, especially in the region beneath the anterior spine. Any part of the appendages that cannot be folded underneath the carapace will similarly only extend slightly into the slow moving flow immediately around the body and thus would not bias the analyses.

The authors do make some comparisons to drag coefficients of species *G. ingens*, which is appreciated. However, more evidence is needed for the role of the third dimension, perhaps with some test cases with three dimensional simulations. They are using 3D simulation software after all and not theory which generally favors 2D.

As outlined above, flow is not three dimensional at the Reynolds numbers of interest in our study. To confirm this, we have run a three dimensional test case for *Isoxys acutangulus*. We provide details of this in the new Supplementary Materials 4. This 3D analysis shows that there is no 3D nature to the wake, and that planes cut through the 3D analysis are equivalent to what is shown by the 2D analysis. Thus a 2D analysis is preferable for our analyses, as they allow a higher resolution mesh for a lower computing cost.

Referee 2 cast doubt on whether a 2D simulation was appropriate, and suggested that *Isoxys* may have swam with its valves spread in ‘butterfly’ position. In a detailed response to this comment (below) we outline why this referee is mistaken – there is fossil evidence from numerous *Isoxys* species from multiple deposits that *Isoxys* had a narrow profile, and was not able to alter the angle between its valves. The ‘butterfly’ preservation of *Isoxys* in the fossil record are considered to represent exuviae. Thus, for an animal with a narrow profile that generates a steady wake, 2D simulations are appropriate.

Overall, Reviewer 2 need more evidence that *Isoxys* is a strong enough swimmer to traverse large distances. The authors give lift and drag coefficients, but the swimming speed and migration ability needs to be made more clear. More hydrodynamic justification and more evidence from extant species is needed to prove this authors' hypothesis.

We have made more explicit our discussion of the corroborating evidence from the fossil record that more streamlined *Isoxys* appear better adapted for swimming than those that lived close to the seafloor, as benthic or possibly hyperbenthic animals (fourth paragraph in Discussion). We have provided hydrodynamic justification for focusing on the carapace (and excluding the eyes, which have a minimal impact – Supplementary Materials 3), while the fossil evidence contradicts the hypothesis that *Isoxys* was able to change the angle between the valves (response to Referee 2 below, sentence added to Materials and Methods paragraph 1). The work we have done shows that our comparison with *Gnathophausia zoea* is appropriate, and provides strong evidence that some *Isoxys* species were capable of moving vertically in the oceans during the Cambrian.

Reviewer 3 also suggests reduction of the claims in the title, in particular that *Isoxys* was the first biological pump. They also request consideration of the influence of oxygen. Please take into consideration comments of all reviewers in your revision.

We have changed the title and incorporated oxygen more fully into the discussion (detailed below in responses to Referee 3).

Reviewer(s)' Comments to Author:

Referee: 1

Comments to the Author(s)

This paper represents a very valuable contribution to our knowledge of the origin and evolution of the pelagic communities in the Early Palaeozoic and it provides for the first time data suggesting the presence of vertically migrating zooplankton in Early Cambrian oceans. The paper is well written and figures/supplementary information are of very good quality and thus the paper could be published as it is.

We thank the referee for their positive comments. To read that the paper could be 'published as it is' was very nice. We have considered the suggestions below to improve the manuscript in detail, and while we cannot follow them all we thank the referee for the suggestions.

I just have a few questions/suggestions that may or may not enter the scope of this paper and that motivated my recommendation to "accept with minor revisions":

- Recent planktonic organism (including the *Gnathophausia* species) and other Palaeozoic supposedly planktonic taxa (e.g. graptolites, conodonts, myodocope ostracods...), have a cosmopolitan/transoceanic distribution. How can you explain that it is not the case for (at least some) *Isoxys* species? Is that related to taphonomic processes and/or the lack of Lagerstätten in which they could be preserved? A palaeobiogeographic distribution map of the mentioned *Isoxys* species and a recent distribution map of *Gnathophausia* species would be valuable additions to the paper.

The lack of individual *Isoxys* species with a global distribution has been noted before (e.g. Stein 2010), and the referee is right to raise this as a point to consider. The referee has also identified the reason why we think this is likely the case – taphonomic process and Lagerstätten distribution in time and space.

In our study, we identify two taxa which are particularly adept swimmers that we expect to have the widest distributions: *Isoxys longissimus* and *Isoxys paradoxus*. The *Isoxys* species with higher drag coefficients and lower lift coefficients we expect to be provincial. However, *Isoxys longissimus* and *Isoxys paradoxus* are also the rarest *Isoxys* species in the Burgess Shale and Chengjiang respectively, a result of taphonomic processes (as discussed in the main text, living far from the sea floor reduces preservation potential). Thus, it is only in deposits which preserve a large quantity of soft bodied material, such as the Tier 1 (*sensu* Gaines 2014) Lagerstätten that we preserve evidence of these vertical migrators. This is compounded further by looking at the depths at which *Gnathophausia* have been collected (Meland & Aas 2013, *A taxonomical review of the Gnathophausia (Crustacea, Lophogastrida), with new records from the northern mid-Atlantic ridge*). Most are collected at depths >1000 meters, well below the depth of Cambrian deposits preserving soft-bodied material.

The issue is further complicated by the lack of coeval Lagerstätten – given the difference in ages, we would not expect the same species in the Burgess Shale and Chengjiang. Unfortunately we lack Tier 1 Lagerstätten of the same age from multiple palaeocontinents at this moment in time.

Thus, sadly, a comparison between *Gnathophausia* species in the modern and the geographic ranges of *Isoxys* species in the fossil record would not be a fair comparison in our opinion.

We do discuss the case of *Isoxys longissimus* in Utah, from the Wheeler Formation (slightly younger than the Burgess Shale). This occurrence is interesting and adds weight to a wider geographic distribution for this species than *Isoxys acutangulus* (also from the Burgess Shale but not known from Utah). *Isoxys acutangulus* is so much more abundant than *I. longissimus* in the Burgess Shale, and so, should the two occupy a similar ecological niche and water depth, we would expect to find *Isoxys acutangulus* and not *Isoxys longissimus* with a wider distribution, as a function of abundance.

We have expanded this part of the discussion to take into account the issues raised by the Referee, it now reads: '*Isoxys* species are not globally distributed [42]. Many species (for example those clustering with *Surusicaris*) appear suited to hyperbenthic habits, and so would be expected to have provincial distributions. The limited geographic distribution of *I. longissimus* and *I. paradoxus* is most likely due to a combination of factors. Firstly, deposits where *Isoxys* is expected to be preserved are not evenly distributed in time and space – Stage 3 deposits are mostly in South China, while Wuliuan and younger are mostly in Laurentia [43], though the absence of the Chengjiang species *I. paradoxus* in Sirius Passet is notable. Secondly, the lower preservation potential of pelagic (compared to hyperbenthic) species means that they are rare even in Tier 1 Burgess Shale-type Lagerstätten (*sensu* [43]). However, despite its rarity, the Burgess Shale species *I. longissimus* has a wider known geographic range than the co-occurring *I. acutangulus*. The former has also been reported from the Wheeler Formation, House Range, Utah, USA [44].'

- Nowhere in the text is mentioned the possible role of the cuticle ornamentation in the hydrodynamics of the carapace. Several species of *Isoxys* and *Tuzoia*, as well as recent planktonic crustaceans, have micro-ornament (e.g. reticulations, corrugations) on their carapace. Is it possible that it played a role in the hydrodynamics of the animal or is it a minor factor compared to the overall shape of the carapace?

Cuticle ornamentation would serve to roughen the surface. Roughened surfaces in some specific cases (such as the dimples in golf balls) have been shown to reduce drag, however this is at much higher Reynolds numbers than applicable to *Isoxys* in this study. Vogel in *Life in Moving Fluids* provides a good overview of how this works (pages 99–101 in the paperback 2nd edition): 'Usually, and the point should be emphasized, roughness is either without consequence or it increases drag. At low Reynolds numbers small bumps will be within the slowly moving fluid near the surface and be of little consequence.' The special case of the dimples in the golf ball helps by promoting turbulence near the surface of the object, and thus postponing separation as the fluid travels around the bluff body. This occurs (for a golf ball) at Reynolds numbers ~50000–150000.

We have added a sentence in the Materials and Methods to make this explicit – that we do not expect the cuticle ornamentation to be significant. 'The cuticle ornamentation in some *Isoxys* species is not expected to impact the drag at the low Reynolds numbers considered in this study, as the roughened surface falls within the slowly moving fluid near the carapace surface [32].'

- Could there be an impact of the position of the swimming appendages on the hydrodynamics of the animal? *Gnathophausia* have its appendages outside its cephalic shield (a bit like a propeller pushing the streamlined front of the animal). *Isoxys*, on the other hand, has its appendages inside/below the bivalve carapace, wouldn't that be a problem for vertical migration?

The fluid dynamics simulations assume that the animal was swimming horizontally, so the thrust provided by the appendages (either the 'propeller' or *Gnathophausia* or the swimming flaps of *Isoxys*) is in the horizontal plane (x axis of the simulation). The lift (y axis of the simulation) is generated by the carapace. An analogy is the engines and wing of an aeroplane. The engines provide horizontal thrust (x axis), the wings generate lift (y axis). Thus the location of the appendages which generate the horizontal movement would not prove problematic for vertical migration. A sentence mentioning that the propulsion derives from appendage movement has been added to the "Computational Fluid Dynamics" section of the Materials and Methods.

- Maybe an interesting additional reference regarding putative planktonic Bradoriids: Williams, M., Vandenbroucke, T. R., Perrier, V., Siveter, D. J., & Servais, T. (2015). A link in the chain of the Cambrian zooplankton: bradoriid arthropods invade the water column. *Geological Magazine*, 152(5), 923-934.

We have altered our wording about bradoriids, to reflect that most of this group are not considered pelagic. We wanted to remark upon the oldest evidence for millimetric euarthropods in the SSF record at the Stage 3 boundary, as euarthropods are such common elements in modern plankton, and were likely among the first to employ a pelagic and vertically mobile lifestyle. We also now include a reference to the specific case of *Anabaroichilina* and how this relates to the likely strengthening of the biological pump through the Cambrian (linking to a point made by Referee 3). This part of the paragraph now reads: 'Euarthropods, likely early occupants of the plankton [36], are represented in the SSF record from the Cambrian Stage 3 by millimetre scale bradoriids and others (Fig. 5) [e.g. 46,55,56]. The first macroscopic nektonic suspension feeders, such as the radiodont *Tamisiocaris*, also appear at this time [57], while the first centimetre-scale phytoplanktivores are identified close to the Stage 3–4 boundary (Fig. 5) [12]. These data suggest that there was an increase in the diversity of millimetre-scale zooplankton at or close to the base of Stage 3, very close in time to the appearance of the first vertical migrants. Most bradoriids are considered benthic, however *Anabaroichilina* increased its distribution in three phases, providing complementary evidence for a steady strengthening of the pump during the Cambrian. In Epoch 2 *Anabaroichilina* was coupled with benthic assemblages, by the Wuliuan it spread to a wider spectrum of lithofacies, and by the Guzhangian two species became widely distributed [47]'

Referee: 2

Comments to the Author(s)

By employing Morphometric analyses and by modelling the hydrodynamic behaviors of the carapaces of various Cambrian arthropods *Isoxys* spp., Pates et al. explore the multiple morphospaces that the different species of *Isoxys* occupied, and discuss their capability/probability of migrating vertically in the Cambrian ocean. They argue that these bivalved arthropods conducted active migration in the water column. They concluded that by doing this *Isoxys* and the related/similar arthropods could have connected the oceanic sub-ecosystems at different fathoms, which is analogous to the so-called “biological pump” described from modern ocean.

This study is innovative, not only in the methods it utilizes, but as well in the perspective it explores the palaeo-ocean. And the topics it discusses, i.e., the role of a widely distributed arthropod group in the early—middle Cambrian oceanic ecosystem, and the structure and function of such early ecosystem on Earth in comparison to that of today, is potentially of scientific importance. However, I cannot recommend publication of this manuscript in Royal Society, for the following major concerns:

We thank Referee 2 for taking the time to thoroughly go through our manuscript, and for the kind words that this work is innovative and of scientific importance. The points raised have served to highlight areas where we need to clarify particular arguments in the text to address possible misconceptions and misunderstandings. We disagree that the issues raised impact our conclusions or the validity of the findings, and in the point by point discussion below we outline where we have made changes in the text in response to the Referee’s comments. We thank the Referee for engaging with this project, and hope that our answers assuage their concerns.

1) The presence of biological pump in the Cambrian period is based on weak, limited evidence. The authors conducted hydrodynamic simulation on only the carapace. The swimming capacity, however, does not only depend on its carapace design, but probably more importantly on appendage organization. The latter has been totally ignored.

We have demonstrated that the carapace shape significantly influences the drag and lift generated at different Reynolds numbers (i.e. across a range of swimming speeds and body sizes) and have identified the morphological features on the carapace that reduced drag and increased lift. This unequivocally demonstrates the importance of the carapace shape for swimming capacity.

For the purposes of the fluid dynamics simulations, we assumed that the swimming speed was the same for all taxa. This swimming speed was based off a modern analogue, the euarthropod *Gnathophausia*, which is a similar size to *Isoxys*. This is a standard approach for understanding swimming capabilities in extinct animals. For example, a recent study published in *Proceedings B* by Gutarra et al. (2019) exploring the hydrodynamics of ichthyosaurs used observed swimming speeds in extant tetrapods of similar dimensions.

In the discussion we do consider the appendages, specifically comparing two Chengjiang taxa, *Isoxys auritus* and *Isoxys curvirostratus*, whose soft parts were described in detail by Fu et al. (2011). *Isoxys auritus*, which our fluid dynamics simulations demonstrate generates higher drag and lower lift coefficients than *Isoxys curvirostratus* (and thus we determine is a less adept swimmer), also displays appendages with robust endopods more suited to interacting with the sediment. *Isoxys curvirostratus* displays strong swimming flaps and slender endopods, indicative of a free swimming lifestyle. Through considering the appendages, their organisation and morphology, our argument is strengthened. However, Referee 2’s point indicates that we were

not clear enough in explaining this, and so we have altered the wording in this paragraph to make the connection more explicit. The sentence now reads: ‘The stout endopods of *I. auritus* appear well-suited for interacting with the substrate, while exopods with broad fringing lamellae and a sophisticated vascular system in the more streamlined *I. curvirostratus* suggest it was a more powerful swimmer, providing additional support for a pelagic habit.’

Also, I’m not sure about the credit of the hydrodynamic modulation. For one thing, it is difficult to know the exact width of the carapace due to significant compression in the Burgess-type fossils. This would significantly bias the analysis.

Numerous *Isoxys* species preserved in dorsoventral and/or oblique compressions, from multiple Lagerstätten, demonstrate a slender gape and narrow profile.

These include: *Isoxys acutangulus* and *Isoxys longissimus* (Burgess Shale, Garcia-Bellido et al. 2009a figures 5 and 7); *Isoxys communis* (Emu Bay Shale, Garcia-Bellido et al. 2009b plate 1, figure 7); *Isoxys curvirostratus* (Chengjiang, Fu et al. 2011 figure 3B); *Isoxys minor* (Guanshan, Wang et al. 2012 figure 2e); *Isoxys shandongensis* (Mantou Formation, Wang et al. 2010 figure 2e).

We have increased the number of references to support the statement in the Materials and Methods section (subheading Computational Fluid Dynamics) ‘undeformed *Isoxys* specimens preserved in dorsal view show a narrow profile’ to emphasise the quantity of evidence that *Isoxys* had a narrow profile in life.

Meanwhile, the three-dimensional shapes of the animals’ body cannot be determined; probably they are not fixed: the animals could have adjusted the angle between two valves during swimming to obtain different hydrodynamic performances. Please imagine extreme occasions where *Isoxys* swims in a “butterfly” position, or with their valves only slightly open. The lift, drag forces would be completely different.

The fossil record does not provide evidence to support the Referee’s hypothesis – that *Isoxys* could adjust the angle between the valves during swimming. In fact, the current fossil evidence suggests the opposite – that *Isoxys* spp. could not move the valves. No adductor muscles have been reported (despite abundant specimens preserving soft parts), and so there is no muscular mechanism by which *Isoxys* spp. could have adjusted its carapace gape. Indeed Vannier & Chen (2000) state: ‘nothing indicates that *Isoxys* was able to control the ventral gape of its carapace’. Numerous dorso-ventrally compressed specimens (listed above) demonstrate that *Isoxys* had a narrow gape. All *Isoxys* specimens preserving soft parts (with one exception – Stein et al. 2010) are preserved dorso-ventrally compressed, or in lateral view with one valve compressed over the other, suggesting that these are carcasses and reflect the morphology of the animal in life. Meanwhile, specimens in ‘butterfly’ orientation are generally considered exuviae (Garcia-Bellido et al. 2009b).

To clarify this, we have added sentences immediately following the statement that *Isoxys* had a narrow profile in life (above) to state directly that *Isoxys* was unable to alter the angle between the valves during swimming, and that butterfly specimens are most likely exuviae, with citations from the literature. The new sentences read: ‘Propulsion during swimming derived from movement of the ventral appendages, and not from flapping of the bivalve carapace [15, 27]. A lack of adductor muscles means that *Isoxys* was unable to alter the size of the gape during swimming [15], and the numerous specimens preserved in ‘butterfly’ orientation are considered exuviae [27]’

In addition, although the authors claimed that the effect of the eyes can be ignored, this is incredible given the sizes of the spherical eyes of *Isoxys*. The great appendages in a number of *Isoxys* species would have important effects on swimming too. All in all, it is not acceptable to base the hydrodynamic result on the compressed (deformed) carapace alone. As suggestions, I would like the authors to include at least some soft-part data in their analyses, for example the eyes and the great appendages that significantly protrude beyond the carapace. I know this would be difficult, but the authors can choose to focus on a few species and perform more comprehensive analysis (e.g., considering different compression ratios, and different angles between valves while swimming, etc), rather than dabble many species.

We did not include eyes in the analyses in our original submission because we were interested in features under hydrodynamic control like the carapace and not those under different selective pressures, such as the eyes. When determining the adaptations of different species to different lifestyles, we felt it was more appropriate to focus on features that are likely to be selected for, and where we can link form to function.

However, we accept that it is plausible that soft parts that extruded beyond the carapace (such as eyes) could possibly impact the hydrodynamic performance of *Isoxys* taxa in terms of the comparison with *Gnathophausia*. To test this we conducted 66 additional simulations for two taxa, *Isoxys acutangulus* and *I. longissimus*, that include the eyes in two positions. The results demonstrate that the eyes had minimal impact on the drag and lift coefficients, and thus had minimal impact on the hydrodynamics of the organisms. We have included details of this analysis in Supplementary Materials 3 and added sentences to the Materials and Methods and Results sections which briefly report on this additional set of analyses.

These analyses considering the eyes demonstrate that small protrusions of soft parts at the anterior of the carapace (below the anterior spine) have minimal impact, as the water flow at this point is slow (see short discussion in Supplementary Materials 3). We do not expect the raptorial appendages to protrude beyond the carapace (as these are flexible they could have been tucked underneath during swimming), but even if they did protrude slightly, we can be confident that the impact on drag and lift would be minimal, just as for the eyes.

As outlined above, the current fossil evidence indicates that *Isoxys* did not adjust the angle between its valves as it swam, and a narrow gape has been observed in multiple taxa. Thus we choose not to interrogate the hypothesis that *Isoxys* adjusted the angle between its valves.

2) If the authors want to argue about the vertical migration of *Isoxys*, they should not only explore the hydraulic design of the carapace and the propulsion of the appendages, but should also seek for fossil evidence that these animals are preserved/living at different environmental settings, in particular different depths of the water column.

Our study not only explores the hydrodynamic role of the carapace, but also uses evidence in the fossil record such as soft parts and relative abundance in deposits. The relative abundance in different deposits provides evidence that different species were living at different depths – some close to the seafloor and others in the water column. We have edited the fourth paragraph of the Discussion (beginning ‘Corroborating evidence for variation in bathymetric range for different *Isoxys* species comes from the fossil record itself’) to make this argument more explicit.

As the vast majority of deposits preserving soft tissues represent very similar depositional settings (the Guanshan and possibly the Emu Bay Shale are exceptions), this evidence comes from the

relative proportions of each *Isoxys* species in each deposit. For example in the Chengjiang *Isoxys auritus* is by far the most abundant, and *Isoxys paradoxus* the rarest. As *Isoxys auritus* is inferred to have lived close to the sediment (the hydrodynamic performance of its carapace is poor compared to the other *Isoxys* taxa from Chengjiang) it is far more likely to be preserved than *Isoxys paradoxus* which is inferred to be vertically mobile. Similar patterns are identified in other Lagerstätten (e.g. Burgess Shale). Further evidence is provided by the greater biogeographic range of *I. longissimus* compared to *I. acutangulus* despite the latter's far greater abundance in the Burgess Shale.

We have edited this paragraph to make more explicit the link between inferred life habit and preservation abundance. For example we have added the sentence 'In general, species with inferred vertically migrating lifestyles are much rarer than those that lived close to the seafloor.' Immediately before the section where we go through individual deposits species by species.

3) The authors hypothesized that *Isoxys* species (and other similar bivalved forms) occupied different niches. This is based mainly on the morphometric analysis. However, shape variation occurs in many animal groups and can be the results of various adaptations. Therefore, shape variation can serve as only supplementary evidence for niche diversification. Again, I think soft anatomy would be (more) important. There are rich literatures describing the morphologies of the eyes and the great appendages, both of which are crucial for feeding and movement of *Isoxys*.

We combine the morphometric analysis with the fluid dynamics simulations to demonstrate that the morphological variation observed impacts hydrodynamic performance, and thus niche occupation. As discussed above, we have identified the parts of the carapace that influence the hydrodynamics (i.e. long spines generate lift, asymmetry and more slender shape reduces drag). Thus our conclusions are based on more than the morphometric analysis, and we have linked form to function using the fluid dynamics experiments.

As discussed above, the soft parts of *Isoxys* provide further support for the conclusions drawn from the morphometric analyses and fluid dynamics simulations. This is discussed in the main text, the fifth paragraph of the Discussion, beginning 'Further support for the Chengjiang taxon *Isoxys auritus* occupying a niche closer to the seafloor than *I. curvirostratus* comes from a comparison of the soft anatomy (soft parts are unknown in *I. paradoxus*).'

4) The discussions in section "Metazoans and the Cambrian biological pump" are generally good, but not well organized. I'm happy to see SSF is considered, but in general this section seems to have deviated from the main points of the manuscript, and have been based on poor evidence – too ambitiously covered many taxa, Lagerstätten and ecological concepts.

We have added some clarifications and reworded parts of this section following this comment and more specific suggestions from the other referees. Figure 5 has also been edited to include more details. We feel that this section is important and want to retain it in the manuscript, as we need to place our findings into the bigger picture.

5) From the carapace design and known appendage structures of *Isoxys*, I believe it can swim. But I feel a bit reluctant to think that it can migrate across great depths. There are many other arthropods in Cambrian Lagerstätten that, judged from their body design, seem to be more adaptive to swimming, such as *Tokummia* and *Waptia*. As to me it is more favorable to explore the migrating

capability of these animals.

We appreciate that the Referee agrees that *Isoxys* was able to swim. We have demonstrated in our study that some species had more hydrodynamic carapaces, and thus were likely better swimmers than others. We have also demonstrated that some possessed carapaces more suited to generating lift than others. We would also like to emphasize that we are not saying that all *Isoxys* were vertical migrators, in fact our data suggest the opposite – that most would have remained close to the seafloor. Two taxa in particular display adaptations to moving vertically in the water column, providing data that they were able to migrate across depths. This is unusual for Cambrian animals, and important for understanding the palaeo-ocean. This is why we chose to focus the discussion on the importance of the few vertically mobile *Isoxys* species for our understanding of the Cambrian biological pump.

We have plans to use a similar approach to interrogate the hydrodynamics and swimming capabilities of other Cambrian euarthropods, such as those suggested by Reviewer 2, in future projects. If the results show that these taxa were also able to migrate vertically in the water column, that would be an exciting result and add to the known complexity of the Cambrian biological pump yet further. However, such a project will be a large undertaking and is beyond the scope of our current study, which is focused on *Isoxys*. Over time and a number of projects we hope to build on this study to provide a more complete understanding of the swimming capabilities of Cambrian euarthropods.

Other comments are detailed in the annotated pdf. I hope these could help.

Line 45 – carcasses are important but not shown in Fig. 1. – **we thank the Referee for pointing out this oversight, and have added carcasses to Fig. 1.**

Line 96 – I would expect that the hydrodynamic performance of the carapace is sensitive to its thickness. Although you may use the dorsally preserved specimen to justify the thickness, I'm wondering whether such data are representative of the true thickness of the animal in vivo. The two valves can have different angles when living (through which the animal might have adjust its lift force). Imagine the difference between a "butterfly" position and a tightly closed position of the two valves in swimming. – **this has been addressed above.**

Line 101 - This is not the excuse for excluding them [the eyes]. Eyes and the frontal appendages protrude beyond the carapace. Other appendages have been described in previous literatures (and you have drawn a simplified model in Fig. 2). Because they're big they probably had significant effect on the drag/lift force. Excluding them would significantly biased the result. – **this point was also made above, we have run analyses including the eyes which demonstrate minimal impact of soft parts which are not completely covered by the carapace.**

Line 112 - Did you model the small juveniles and the adults separately? They are different in both shape and size. – **in this study we are focused on the adults, and so used adult morphology throughout. A detailed study on the hydrodynamics of the juveniles would include much lower Reynolds numbers, and is beyond the scope of this study. It is planned as a follow up.**

Line 153 - While "deep carapace" is not surprising because the outline analysis includes this feature, "short spines" are features independent of the outline. These two should be mentioned separately.

– **We have reworded the sentence, which now reads ‘Species clustered with *Surusicaris* have symmetric and deep carapaces *and* relatively short spines.’**

Line 158 - As long as I see, this simulation has two defects: 1. The true thickness (width) of the animals is unknown. 2. Propulsion of the appendages and their hydrodynamic performance not known. Without such information the present results would be largely biased. – **these points have both been addressed above. A 2D simulation is appropriate because of the steady state laminar flow and narrow carapace (which is well known in the fossil record). A modern analogue has been used for the swimming speed, which is a common approach in studies of hydrodynamics on animals known from the fossil record.**

Line 176 - Either morphometric analysis or hydrodynamic simulation, or both, can only lend limited support to the vertical migration of these animals. Other driving forces for such a behavior, such as searching for food, light, and oxygen, and escaping from predators, are not explored/discussed. These factors could easily mask the capability of migrating vertically provided by the carapace morphology. In addition, the swimming ability of an organism would depend to a large extent on its appendage design, which is not discussed here. – **The Referee lists ecological reasons (light, food, oxygen, predators) why animals may vertically migrate, however these are not testable with fossils. We chose to focus on the carapace morphology, where we can link form and function. We do discuss the appendage design, comparing *Isoxys auritus* and *Isoxys curvirostratus* from the Chengjiang biota (as discussed above).**

Line 178 - Too weak evidence [for occupation of multiple niches]. More needed from appendage structure, carapace composition, and preservational settings of the fossils (which could provide information for their living environment). – **we compare appendage structure of co-occurring species in the Chengjiang biota. We also compare the abundances of different taxa in the same deposits. We have assumed that the carapace composition of all species was similar. We have added ‘All *Isoxys* species were assumed to have the same carapace composition and density’ to the methods to make this assumption explicit.**

Line 182 - How much confidence is there in the data, without true thickness of the animals and the angles between the two valves being known? – **Evidence from numerous species and deposits strongly suggests that *Isoxys* had a narrow profile (as discussed above).**

Figure 1 – You may add "shallow ocean" above the dashed line, corresponding to "deep ocean". – **we have separated this figure by ‘euphotic zone’ and ‘deep ocean’.**

Figure 2 - Genus and species names need to be in italics. Check other figures. I'm sorry but the photographs here are of poor quality. The outlines are not well defined.– **the lack of italics is a result of the online upload system and will be correct in the final version. We have tweaked the colour profile of the photographs in Glimpse to make the outlines clearer, and are happy to work with the editorial team to make them as good as possible. Unfortunately one of us (SP) had planned to photograph specimens specifically for this project, but due to the Covid-19 pandemic was unable to access material, so we are reliant on images that the museums have kindly made available for use.**

I suggest the authors to focus on only a few taxa, and to base the ecological conclusion on both carapace hydrodynamics and feeding and locomotory merits of the eyes and the frontal appendages of *Isoxys*, thus making the analyses more in-depth and convincing. It is also important to seek for fossil evidence for the presence of *Isoxys* in different depths of the Cambrian ocean. I agree that

Isoxys and *Tuzoia* are important bivalved arthropods of the Cambrian, and should have played some role. But they may not be the best organisms for testing the biological pump hypothesis. Moreover as mentioned above, the present evidence is too weak for vertical migration.

We have worked hard to assuage the Referee's concerns about the 2D nature of the analyses, and demonstrated the minimal impact of features which extend beyond the carapace. While we appreciate that the Referee would have preferred a different set of study animals, we feel that we have demonstrated that some *Isoxys* are adapted for a pelagic lifestyle and moving vertically in the water column. This does not mean that other Cambrian animals suggested by the Referee are not also important and relevant. We would like to emphasise that we have already covered the topics raised by the Referee in the text, and have worked to rephrase these sections to make their inclusion more clear.

Referee: 3

Comments to the Author(s)

Dear Stephen, Allie, David and Imran,

this is a very nice and interesting paper, congratulations!

The paper is well written and a suitable contribution for the journal.

Thank you Christian!

Here are a few things, which I think you can improve easily:

1. The title suggests that *Isoxys* alone made the first biological pump. I know, *Isoxys* is not super-rare and all, but it is also not so overwhelmingly abundant that you would expect it to revolutionize all the world's oceans of the time. I would formulate it a bit more humbly.

We have changed the title to reflect this important point.

2. You discuss this later, but for me, to make a point that *Isoxys* was the first engineer of the biological pump, I would expect some evidence that they had a significant biomass that was migrating vertically. I would at least shortly discuss this in the methods, even if it is mentioning that they are not so super-common but that you think that this is due to a taphonomic bias, making the preservation of pelagic plankton less likely (with references).

This is an important point, and we have chosen to add it to the discussion (section Metazoans and the Cambrian biological pump). We have added a sentence: 'For this vector to be significant by the Cambrian Stage 3, a large biomass of *Isoxys* would need to move vertically. While pelagic animals have a lower preservation potential than benthic ones (for example very few fossil copepods are known [49]), *Isoxys* species with inferred (hyper)benthic habits are extremely abundant in both the Chengjiang and Burgess Shale [15,16,21,37–39]), suggesting that their pelagic counterparts *Isoxys longissimus* and *I. paradoxus* may have been similarly numerous.'

3. You start the discussion with mentioning that *Isoxys* was not benthic. Please shortly state WHY people think this is the case.

We have added text on why people have suggested that *Isoxys* was not benthic to the start of this part of the discussion (subheading Vertical migrations and niche partitioning in *Isoxys*). This sentence now reads: 'Functional morphology of *Isoxys* fossil specimens supports an off-bottom (hyperbenthic or pelagic) life habit for this animal[15,16,18,19,35], based on the eye orientation (forwards, slightly ventral) and the elongate slender carapace shape of *Isoxys*.'

4. I recommend to use 'migrant' uniformly instead of switching between migrator and migrant. Migrator sounds strange in my ears.

We have altered the text and now use 'migrant' uniformly as suggested.

5. I would point out that the biological pump probably started out weakly and became stronger through the Phanerozoic. In a way, you do, but it gets a bit lost in details. Maybe you could try structuring the discussion a bit better.

This is an important point to raise – we have added a sentence at the end of the first paragraph of this part of the discussion to make this point explicitly: ‘A series of metazoan innovations which appear in the fossil record in quick succession during the early Cambrian gave the biological pump a modern-structure (Fig. 5) which was strengthened during the Phanerozoic.’ We have also added a statement in the penultimate paragraph that additional metazoan innovations during the Palaeozoic would have further strengthened the pump: ‘Indeed it [the biological pump] likely strengthened through the Palaeozoic with an increase in biomass (from an increased number of taxa, individuals, and size of individual).’

6. I think the transport of oxygen is equally important to the nutrients, but oxygen is rarely mentioned. Maybe stress this a bit more. See Butterfield's paper on the pelagic bioturbation.

We have made mentions of oxygen and oxygenation of deeper waters more prevalent in the discussion, including additional references to Butterfield (2018) *Oxygen, animals and aquatic bioturbation: An updated account*.

7. The resemblance to thylacocephalans and phyllocarids is remarkable. Maybe it would be worthwhile discussing similarities shortly and mentioning that this kind morphology evolved convergently several times? Such an ecomorphological comparison might also help to strengthen some of your points.

We acknowledge that there are numerous arthropods with large carapaces that cover much of the body throughout the Phanerozoic rock record, including groups such as the thylacocephalans and phyllocarids. For the former, the mode of life is highly debated, with suggestions ranging from infaunal (Briggs & Rolfe 1983) to sessile (Alessandrello et al. 1991) to benthic (Pinna et al. 1982) to swimming (Vannier et al. 2006; Schram et al. 1999; Rolfe 1985), with some suggesting they were too large to swim (Secretan 1985). For the latter, only one living pelagic phyllocaridid is known, with the rest being interpreted as benthic (Brahm & Geiger 1966). While we agree that this broad morphology evolved several times, we find that these two examples don't strengthen our points about vertical migrations, because their mode of life is rather different to what our analyses show for *Isoxys*. The morphology of their carapaces is only broadly similar to that of *Isoxys*, and in details they are rather different, perhaps reflecting that there are a wide variety of different ecological pressures that may have influenced the evolution of these two groups (e.g. protection from predation; brooding chamber; feeding, etc.). As such, we chose not to add mention of this point into the manuscript, because we find it outside the scope of our study.

I made some more remarks in the annotated pdf.

Annotations in the pdf:

Line 1 – we changed the title as suggested (comment 1 of main suggestions above)

Line 71 – we altered the order as suggested

Line 181 – we replaced ‘bathymetric’ with ‘depth’ as suggested.

Line 217, 255 – typos, all corrected, thank you for highlighting these.

Line 277 – we rephrased as suggested (replaced ‘building block’ with ‘step towards’)

Line 294, 296 – replaced ‘migrators’ with ‘migrants’ as suggested above.

Further annotations:

Line 23 – likely for early cephalopods as well. And we don't know what all sorts of larvae did... - **we agree that early cephalopods and some larvae likely moved vertically in the water column. However to our knowledge no quantitative work like the fluid dynamics simulations in our study have been done. Thus we feel that our qualifier – ‘first quantitative evidence’ is ok.**

Line 73 – To be relevant for the biological pump, there would have to be many individuals. Is there a way of estimating their abundance? I mean, they were not rare, but is there something (semi-)quantitative? Maybe mention the countries/ regions where they occur to make a point about abundance and the importance of its ecological role. - **This is an important point to raise, and we have chosen to address it in the Discussion. We have added the following, in the first paragraph under subheading: *Metazoans and the Cambrian biological pump*: ‘For this vector to be significant by the Cambrian Stage 3, a large biomass of *Isoxys* would need to move vertically. While pelagic animals have a lower preservation potential than benthic ones (for example very few fossil copepods are known [49]), *Isoxys* species with inferred (hyper)benthic habits are extremely abundant in both the Chengjiang and Burgess Shale [15,16,21,37–39]), suggesting that their pelagic counterparts *Isoxys longissimus* and *I. paradoxus* may have been similarly numerous.’**

Line 178 – orientation of eyes could provide supporting evidence for living off-bottom. – **we have provided a brief explanation for why *Isoxys* is considered to have lived off-bottom, including a reference to the eye orientation. This sentence now reads: ‘The combination of the eye orientation (forwards, slightly ventral) and elongate slender carapace shape of *Isoxys*, supports an off-bottom (hyperbenthic or pelagic) life habit for this animal’**

Line 200 – the abdomen of *Gnathophausia zoea* may reduce the added mass in swimming movements, which is dragged behind the moving body. This might play a physical role. – **we agree that the abdomen likely is important in swimming for *G. zoea*, though this study is focused on *Isoxys*. So we have added the clause ‘though the abdomen may also play a physical role’ at the end of this sentence as a qualifier.**

Line 218 – I wonder whether they might have other means to control buoyancy and thus rise and sink? – **this is an interesting item to bring up and consider, however there is no evidence (currently available) from the fossil record that *Isoxys* was able to control its buoyancy. Should this be true it would provide a second mechanism (apart from lift generation with the carapace) by which *Isoxys* could move vertically in the water column, potentially making more taxa important for the vertical migration vector in the biological pump. However, the complementary evidence from the soft body parts (robust endopods in inferred poor swimmers, powerful swimming exopods in inferred good swimmers), as well as relative abundances of inferred vertical migrators (lower than those living close to the seafloor), suggest that if *Isoxys* were able to control their buoyancy and rise and sink, the impact was less than what is provided by the carapace.**

Line 220 – Burgess Shale is deep enough (several 100 m), so pelagic taxa do not have to be ‘off shelf’. – **our intention with this sentence was to communicate that pelagic taxa can live off shelf in the open ocean. We did not mean to imply that they cannot also live over the shelf. We have rephrased the sentence slightly and added a clause for clarification. It now reads: ‘This broader bathymetric range would have included more open water settings, beyond the maximum depth of the shelf where Cambrian deposits preserving soft-bodied fossils occur – *Gnathophausia zoea* for example has been found at depths of up to 3000 metres’**

Line 222 – Not convinced that preservation potential for pelagic taxa are low, provide a reference to support this. Line 222-223 – Not sure of meaning. Do not think pelagic animals live more rarely on the shelf, needs support from a reference. Or do you mean that pelagic animals get more rarely fossilized? – **we intended this sentence to explain why we think that pelagic *Isoxys* are rarer than those which lived closer to the seafloor. The reference supplied in the previous sentence (Allison 1986, *Soft-bodied animals in the fossil record: The role of decay in fragmentation during transport*) is an experimental study demonstrating that euarthropods with carapaces disarticulate after death, with decay leading to greater disarticulation. Thus those with further to travel before burial (pelagic forms) would have a lower preservation potential. We have clarified this sentence to make it more specific to euarthropods (indeed the same reference explains how soft flexible organisms like polychaetes are more resistant to disarticulation than euarthropods), and made the statement comparative ('lower' rather than 'low'). The sentence now reads: 'As modern euarthropod carapaces disarticulate quickly after death [e.g. 40], the preservation potentials for pelagic euarthropods high in the water column are lower than those living closer to the seafloor.'**

Line 234 – sounds also like a taphonomic window. – **We agree that in our interpretation of the Emu Bay Shale, this is likely a taphonomic effect that is influencing the relative abundance of different species of *Isoxys*. We have added "...and creating a taphonomic bias that preferentially preserves pelagic taxa." to the end of this sentence.**

Line 278 – **Our understanding is that 'zooplankton' can be both singular and plural so we have not changed this word.**

Line 291 – the innovation only significant if enough biomass of animals embarked on that mode of life. – **this is an important point to raise, and so we have added the following sentence to provide this context 'The strength of the impact depends on the amount of biomass undertaking vertical migration.'**

Line 297 – not only nutrients are important here – **we agree, and have added 'oxygen and' to this sentence prior to the word 'nutrients' to reflect the importance of vertical migrants in oxygenating the water column as well as transporting nutrients.**

Line 307 – I would suggest that it started weakly and then strengthened with an increasing amount of taxa, number and size as well as metabolism of migrating animals. – **we agree that it likely strengthened through time, and have added a clause at the end of this sentence to that effect, incorporating the points raised here by the referee. We have also added a reference to *Anabarochilina* increasing its distribution through the Cambrian (following suggestions by Referee 1), to further emphasize this point.**

I look forward seeing this paper published!

Best wishes,

Christian

References cited in response to referees

- Alessandrello, A., Arduini, P., Pinna, G. and Teruzzi, G., 1991. New observations on the Thylacocephala (Arthropoda, Crustacea). *In* The early evolution of Metazoa and the significance of problematic taxa (pp. 245-251).
- Allison, P.A., 1986. Soft-bodied animals in the fossil record: The role of decay in fragmentation during transport. *Geology*, 14(12), pp.979-981.
- Brahm, C. and Geiger, S.R., 1966. On the biology of the pelagic crustacean *Nebaliopsis typica* G.O. Sars. *Bulletin of the Southern California Academy of Sciences*, 65, pp. 41–46.
- Butterfield, N. (2017). Oxygen, animals and aquatic bioturbation: an updated account. *Geobiology*, 16, pp. 3-16. <https://doi.org/10.1111/gbi.12267>
- Fu, D.J., Zhang, X.L. and Shu, D.G., 2011. Soft anatomy of the Early Cambrian arthropod *Isoxys curvirostratus* from the Chengjiang biota of South China with a discussion on the origination of great appendages. *Acta Palaeontologica Polonica*, 56(4), pp.843-852.
- Gaines, R.R., 2014. Burgess Shale-type Preservation and its Distribution in Space and Time. *The Paleontological Society Papers*, 20, pp.123-146.
- García-Bellido, D.C., Vannier, J. and Collins, D., 2009. Soft-part preservation in two species of the arthropod *Isoxys* from the middle Cambrian Burgess Shale of British Columbia, Canada. *Acta Palaeontologica Polonica*, 54(4), pp.699-712.
- García-Bellido, D.C., Paterson, J.R., Edgecombe, G.D., Jago, J.B., Gehling, J.G. and Lee, M.S., 2009b. The bivalved arthropods *Isoxys* and *Tuzoia* with soft-part preservation from the Lower Cambrian Emu Bay Shale Lagerstätte (Kangaroo Island, Australia). *Palaeontology*, 52(6), pp.1221-1241.
- Gutarra, S., Moon, B.C., Rahman, I.A., Palmer, C., Lautenschlager, S., Brimacombe, A.J. and Benton, M.J., 2019. Effects of body plan evolution on the hydrodynamic drag and energy requirements of swimming in ichthyosaurs. *Proceedings of the Royal Society B*, 286(1898), p.20182786.
- Meland, K. and Aas, P.Ø., 2013. A taxonomical review of the *Gnathophausia* (Crustacea, Lophogastrida), with new records from the northern mid-Atlantic ridge. *Zootaxa*, 3664, pp.199-225.
- Mittal, R. and Balachandar, S., 1995. Effect of three-dimensionality on the lift and drag of nominally two-dimensional cylinders. *Physics of Fluids*, 7(8), pp.1841-1865.
- Pinna, G., Arduini, P., Pesarini, C. and Teruzzi, G., 1982. Thylacocephala: una nuova classe di crostacei fossili. *Atti della Societa italiana di Scienze naturali e del Museo civico di Storia naturale di Milano*, 123(4), pp.469-482.
- Rolfe, W.I., 1985. Form and function in Thylacocephala, Conchyliocarida and Concavicarida (? Crustacea): a problem of interpretation. *Earth and Environmental Science Transactions of The Royal Society of Edinburgh*, 76(2-3), pp.391-399.
- Schram, F.R., Hof, C.H. and Steeman, F.A., 1999. Thylacocephala (Arthropoda: Crustacea?) from the Cretaceous of Lebanon and implications for thylacocephalan systematics. *Palaeontology*, 42(5), pp.769-797.

- Secretan, S., 1985. Conchyliocarida, a class of fossil crustaceans: relationships to Malacostraca and postulated behaviour. *Earth and Environmental Science Transactions of The Royal Society of Edinburgh*, 76(2-3), pp.381-389.
- Stein, M., Peel, J.S., Siveter, D.J. and Williams, M., 2010. *Isoxys* (Arthropoda) with preserved soft anatomy from the Sirius Passet Lagerstätte, lower Cambrian of North Greenland. *Lethaia*, 43(2), pp.258-265.
- Vannier, J. and Chen, J.Y., 2000. The Early Cambrian colonization of pelagic niches exemplified by *Isoxys* (Arthropoda). *Lethaia*, 33(4), pp.295-311.
- Vannier, J., Chen, J.Y., Huang, D.Y., Charbonnier, S. and Wang, X.Q., 2006. The Early Cambrian origin of thylacocephalan arthropods. *Acta Palaeontologica Polonica*, 51(2).
- Vogel, S., 1996. *Life in Moving Fluids: the physical biology of flow*. Second Edition, revised and expanded (paperback). Princeton University Press. New Jersey, USA. 467 pp.
- Wang, Y., Huang, D. and Lieberman, B.S., 2010. New *Isoxys* (Arthropoda) from the Cambrian Mantou Formation, Shandong Province. *Acta Palaeontologica Sinica*, 49(3), pp.398-406.
- Wang, Y.N., Huang, D.Y., Liu, Q. and Hu, S.X., 2012. *Isoxys* from the Cambrian Guanshan Fauna, Yunnan Province. *Earth Science—Journal of China University of Geoscience*, 37, pp.156-164.